# A hybrid physics–AI approach using universal differential equations with state-dependent neural networks for learnable, regionalizable, spatially distributed hydrological modeling

Ngo Nghi Truyen Huynh[1], Pierre-André Garambois[1], François Colleoni[1], and Jérôme Monnier[2]

[1]INRAE RECOVER, Aix-Marseille Université, 3275 Route Cézanne, 13182 Aix-en-Provence, France
[2]INSA, Institut de Mathématiques de Toulouse (IMT), Université de Toulouse, 31400 Toulouse, France

**Correspondence:** Ngo Nghi Truyen Huynh (ngo-nghi-truyen.huynh@inrae.fr) and Pierre-André Garambois (pierre-andre.garambois@inrae.fr)

**Abstract.** Conceptual hydrological models, traditionally relying on simplified representations of physical processes governed by conservation laws remain widely used in operational hydrology due to their explainability and practical applicability. However, these process-based models inherently face structural uncertainties and a lack of scale-relevant theories—challenges that emerging artificial intelligence (AI) techniques may help address. In parallel, high-resolution models are crucial for predicting extreme events characterized by strong variability and short duration, making spatially distributed hybrid modeling critical in the current context. We introduce a hybrid physics–AI framework that embeds neural networks (NNs) seamlessly into a spatialized, regionalizable, and fully differentiable process-based model via universal differential equations (UDEs). The model integrates a state-dependent NN to refine internal water fluxes and an implicit resolution of the UDE system, followed by kinematic wave routing on a flow direction grid. Spatially distributed parameters are inferred through regionalization mappings including convolutional NNs, and adjoint-based gradients enable end-to-end training of the hybrid system. We implement this framework into the latest release of the `smash` platform, significantly extending its capabilities to comprehensively evaluate hybrid models at kilometric spatial and hourly temporal resolutions. The results show that hybrid approaches demonstrate consistently strong and stable performance in calibration and various validation scenarios. Additionally, the UDE structure exhibits a hybridization effect that modifies state dynamics and runoff flow, achieving more accurate streamflow simulations for flood modeling.

## 1 Introduction

### 1.1 Hydrological modeling and the rise of artificial intelligence

With the explosion of big data and artificial intelligence (AI), research on innovative approaches that leverage the power of AI for flood forecasting and hydrological modeling has demonstrated significant efficiency and advantages in terms of accuracy and computational cost compared to traditional rainfall–runoff models (Sit et al., 2020). Hydrological modeling approaches have evolved from early rational methods that use simple linear equations relating peak discharge to rainfall intensity, through unit hydrograph theory, to complex statistical models that rely on empirical relationships derived from observed data. In paral-

lel, physically-based modeling approaches emerged based on the blueprint model (Freeze and Harlan, 1969), which provides a complete theoretical structure for watershed processes. However, the descriptive equations used for each process require significant simplifying assumptions in real-world applications (Beven, 2002). In the pursuit of robust runoff simulation and/or of physical realism, the "resolution–complexity continuum" (Clark et al., 2017) has been explored over the past five decades with approaches of varied complexity, from black-box to fully physics-based. Conceptual models offer a middle ground, representing watershed processes through simplified conceptualizations that incorporate physical principles. While conceptual models provide more practical implementations than physically-based models and retain more physical interpretability than empirical approaches, they often suffer from several limitations: lack of scale-relevant laws, intrinsic uncertainties in their structure and parameterization, no explicit link between hydrological parameters and physical descriptors, and equifinality in parameter estimation. These limitations have motivated the integration of data-driven modeling approaches, leading to increased interest in applying AI techniques to hydrological modeling (e.g., reviews in Reichstein et al. (2019)).

In general, there are two main approaches for integration of machine learning (ML) and deep learning (DL) techniques in hydrological modeling. The first approach involves using a full black-box model that replaces traditional rainfall–runoff models. These purely data-driven models have been applied successfully to hydrological prediction, achieving state-of-the-art performance in various applications using long short-term memory (LSTM) network (Kratzert et al., 2018; Feng et al., 2020; Cho and Kim, 2022) and their variants like LSTM-based Seq2Seq model (Xiang et al., 2020). For example, Kratzert et al. (2018) reported that across 241 catchments in the U.S., their LSTM model achieved a mean Nash-Sutcliffe efficiency (NSE) of 0.63 in temporal validation, with over 50% of catchments reaching NSE values above 0.65. A second approach is the hybrid method, which seamlessly integrates neural networks (NNs) into process-based numerical models, adhering to the principle of "learning under physical constraints." This strategy aims to harness the predictive power of AI while maintaining the physical consistency and interpretability of hydrological processes within the context of differentiable modeling (Shen et al., 2023). For example, recent advancements in hybrid modeling include leveraging NNs for regionalizing differentiable hydrological models (Feng et al., 2022; Huynh et al., 2024), generating more plausible potential evapotranspiration estimates (Wang et al., 2024), and refining the internal fluxes of rainfall–runoff models (Huynh et al., 2025). Additionally, merging process-based differential equations with ML can be highly advantageous. This has been recently demonstrated with physics-informed neural networks (PINNs) in Raissi et al. (2019), where the process-based model serves as a weak constraint in the training cost function and is well-suited to assimilate observations (e.g., He et al., 2020).

Despite the success of both pure and hybrid ML models, each approach faces several limitations and challenges. First, pure ML models are generally highly sensitive to large and high-quality training datasets, making them less reliable than process-based models in data-scarce regions (Shen, 2018; Reichstein et al., 2019; Beven, 2020; Sit et al., 2020). Unlike physics-based models, which rely on governing equations, ML models do not inherently impose physical constraints, resulting in reduced generalizability under extreme or unseen hydrological conditions (Beven, 2020). Hybrid modeling is a promising approach to address these limitations by objectively integrating ML/DL components into physically based models, rather than replacing them. However, hybrid approaches also face their own set of challenges. A primary limitation is maintaining physical consistency throughout the hybrid framework while leveraging the flexibility of data-driven approaches (Reichstein et al.,

2019). Moreover, the integration of ML components with physical models may introduce complexities in model coupling and error propagation across different components (Frame et al., 2021). Current hybrid approaches are predominantly employed for lumped models, and hybrid modeling at high spatio-temporal resolution has received little attention, likely due to model complexity (with coupled processes) and the challenges of optimizing high-dimensional parameters (with gridded conceptual parameters) over large domains at high-resolution.

## 1.2 Toward a distributed hybrid physics–AI approach with implicit numerical solvers

High-resolution hydrological modeling incorporates spatially distributed information at relatively fine temporal scales, such as gridded radar rainfall, essential for representing extreme flood events characterized by strong variability and short duration. However, such resolutions also introduce challenges for model calibration due to the high dimensionality of optimization parameters. Although this high-dimensional calibration problem can be tackled with a variational data assimilation (VDA) framework using numerical adjoint models of spatially distributed differentiable hydrological models (Castaings et al., 2009; Jay-Allemand et al., 2020; Huynh et al., 2023; Garambois et al., 2025), one is still facing overparameterization issues due to the sparsity of constraining calibration data such as in situ discharge timeseries, compared to large vectors of spatialized parameter. This problem can be addressed by introducing stronger constraints into the forward model, such as learnable mappings between physical descriptors and conceptual parameters (Huynh et al., 2024). In addition, challenges remain due to both input data uncertainty and structural uncertainty in the hydrological model.

Another important point is that the reservoir states in conceptual hydrological models can be computed continuously and simultaneously by numerically solving a system of ordinary differential equations (ODEs) rather than sequentially (e.g., Santos et al. (2018) with state-space GR4 model). Continuous state-space hydrological models (i.e., those formulated within a dynamic system) typically employ implicit numerical ODE solvers to integrate the system's evolution over time. Indeed, under the time-step and forcing conditions tested by Clark and Kavetski (2010), fixed-step explicit schemes produced unacceptably poor hydrological simulations—even for parameter sets yielding good performance—whereas implicit schemes maintained numerical stability and physical consistency. An adjoint-based implicit scheme (designed to simplify the Jacobian computations required for implicit resolution) was proposed for a hybrid lumped model by Song et al. (2024), in which NNs were employed solely for parameter regionalization.

Recent studies indicate that embedding neural networks (NNs) into differential equation-based models, where the NN acts as a functional approximator or correction for selected source terms within a physically based differential equation offers a promising direction for advancing process representation in complex geophysical systems (Rackauckas et al., 2021; Yin et al., 2021; Höge et al., 2022) or biological systems (Philipps et al., 2025). This generic framework, referred to in the scientific ML community as universal differential equations (UDEs, Rackauckas et al. (2021)), provides a flexible approach for integrating prior mechanistic knowledge with data-driven components in differential models. For instance, in hydrological modeling, Höge et al. (2022) implemented NNs that correct or replace precipitation-related source terms without dependence on the model states in the ODE system, and solved the system using an explicit numerical solver. However, to represent physical process memory, feedbacks, and nonlinear state interactions, such NN components should ideally depend on hydrological states, and

the ODEs should be solved using implicit numerical schemes as discussed above. Thus, a rigorous numerical approach for solving UDE system—including NNs that depend on the hydrological model states—using implicit time integration schemes remains unexplored in hydrological modeling. Key challenges include the efficient computation of the Jacobian matrix for state-dependent NNs within the UDE system for its resolution, and the derivation of a numerical adjoint of the complete hydrological model including UDE and gridded kinematic wave (partial differential equation, PDE) routing to enable high-dimensional parameter optimization.

## 1.3 Objectives and contributions of the proposed framework

This study proposes a hybrid physics–AI approach that integrates state-dependent NNs within UDEs, into a differentiable, regionalizable, and spatially distributed hydrological model, where the UDE system is solved using an implicit numerical scheme. With the term "hybrid physics–AI," we refer to an approach that seamlessly integrates NNs directly into the physical differential model and its numerical solver. This differs from approaches that treat NNs separately, for example, as pre- or post-processing components. The approach implements a mathematically rigorous method for computing the Jacobian matrices required by the implicit scheme when incorporating NN components. While we initially focus on simpler NN architectures to ensure numerical stability and physical interpretability, the hybrid numerical solver is designed to be compatible with PyTorch's automatic differentiation (Paszke et al., 2019) and could employ more complex architectures or deeper networks for process-parameterization.

The key novelty of this work lies in its capacity and generalizability to solve a spatialized UDE system with an implicit numerical scheme. This UDE system includes a NN that depends on the model states to refine internal water fluxes (source terms in the right-hand side of the ODEs), which leverages the dynamical system's internal state variables to retain information about past forcings and responses, thereby inducing temporal dependencies analogous to those captured by recurrent neural networks (RNNs). This design allows the model to represent temporal memory effects and learn corrections for structural errors in the conceptual hydrological model. Additionally, the regionalization NN from Huynh et al. (2024) is extended to explore alternative NN architectures, such as convolutional neural networks (CNNs) in addition to multilayer perceptrons (MLPs), to improve the adaptability and scalability of parameter estimation. These developments aim to bridge the gap between ML flexibility and physical model interpretability, uncovering hydrological behaviors and scale-relevant theories inferred with AI techniques. It enables the addressing of several open research questions by providing a robust and powerful tool for enhancing flood modeling, mitigating structural uncertainty in modeling, optimizing data efficiency, and enabling more effective multi-scale information extraction through hybrid flux correction.

This paper also introduces `smash` v1.1, an upgraded version of the `smash` platform, following its initial release v1.0 in Colleoni et al. (2025). The new version includes various hybrid physics–AI hydrological solvers and provides a more comprehensive user guide, along with detailed mathematical descriptions of the implemented models (see smash 1.1.0 Release Notes[1] for details). As the core solver for the French flash flood forecasting system, `smash` is positioned to improve real-

---

[1]https://smash.recover.inrae.fr/release/1.1.0-notes.html (last access: 5 November 2025)

world flood simulation and hydrological forecasting, facilitating the integration of AI-enhanced physics-based modeling into operational hydrology.

## 2 Method

This study employs three key components. First, we implement the continuous state-space GR4 structure presented in Santos et al. (2018) into `smash`, in addition to the classical GR4 model without an explicitly formulated water balance in Perrin et al. (2003). This state-space model solves the water balance differential equations continuously using numerical schemes, instead of splitting the equations to compute the solutions analytically. Second, the seamless regionalization method using NNs, HDA-PR (Hybrid Data Assimilation and Parameter Regionalization), proposed in Huynh et al. (2024), enabling the estimation of conceptual hydrological parameters from physical descriptors, is extended to incorporate CNNs in addition to MLPs. Finally, the process-parameterization NN previously used for the analytical resolution of a discrete state-space model in Huynh et al. (2025)—which relied on an algebraic structure—is now directly embedded into the ODEs. These ODEs are solved using the Newton–Raphson method within an implicit Euler scheme, which mitigates numerical errors that arise when using simple explicit schemes with sequential computations and split operators.

### 2.1 Regionalizable gridded hydrological modeling with UDE and routing PDE

Let us first define the domain, the main quantities of interest, and the general formulation of the dynamic hybrid model along with its dependencies. Consider a two-dimensional spatial domain $\Omega \subset \mathbb{R}^2$, with $x \in \Omega$ as the spatial coordinate and $t \in ]0,T]$ as the physical time. Here, a regular grid is assumed, with an 8-direction (D8) surface flow path drainage network, $\mathcal{D}_\Omega$. The hybrid rainfall–runoff model $\mathcal{M}$ (Equation 1) dynamically maps atmospheric forcings $\mathcal{I}(x,t) = [P,E](x,t)$ onto state variables of surface discharge $Q(x,t)$, internal states $\boldsymbol{h}(x,t)$, and internal fluxes $\boldsymbol{q}(x,t)$, depending on learnable spatio-temporal correction $\boldsymbol{f}_q(x,t)$ applied to internal fluxes $\boldsymbol{q}(x,t)$, on spatialized physical parameters $\boldsymbol{\theta}(x)$ and initial states $\boldsymbol{h}_0(x)$ that can be inferred through learnable regionalization mappings.

$$[Q,\boldsymbol{h},\boldsymbol{q}](x,t) = \mathcal{M}\left(\mathcal{D}_\Omega, \mathcal{I}(x,t); \boldsymbol{f}_q(x,t), [\boldsymbol{\theta},\boldsymbol{h}_0](x)\right). \tag{1}$$

The hybrid spatially distributed hydrological model $\mathcal{M}$ is constructed by chaining differential equations and NNs, consisting of: (i) a dynamic hydrological component $\mathcal{M}_{\mathrm{rr}}$, operating at the pixel scale, that simulates moisture states $\boldsymbol{h}$, internal fluxes $\boldsymbol{q}$, and surface discharge $Q$. This component is formulated as a UDE system, made learnable through a set of NN operators $\phi$ that provide spatialized hydrological parameters $\boldsymbol{\theta}_{\mathrm{rr}}$ and spatio-temporal flux-corrections $\boldsymbol{f}_q$; and (ii) a dynamic and gridded hydraulic routing module $\mathcal{M}_{\mathrm{hy}}$ that transports surface discharge $Q$, whose parameters $\boldsymbol{\theta}_{\mathrm{hy}}$ can also be inferred by a regionalization NN. Together, these components, operating on the same spatial grid, define the forward model that is written, $\forall x \in \Omega, t \in ]0,T]$, as in Equation 2.

$$\mathcal{M} = \mathcal{M}_{\mathrm{rr-hy}}\big(\,\cdot\,; \phi(\,\cdot\,)\big):$$

$$\begin{cases}
\textbf{Hydrological operator } \mathcal{M}_{\mathrm{rr}} \textbf{ (steps (0)–(2))}: \\[2mm]
\textbf{(0) Interception: } \begin{pmatrix} P_{\mathrm{n}} \\ E_{\mathrm{n}} \end{pmatrix} = \mathcal{F}_{\mathrm{itc}}(P, E, c_i) \\[4mm]
\textbf{(1) UDE dynamics (GR4-like):} \\[2mm]
\dfrac{d\boldsymbol{h}}{dt} = \begin{pmatrix} \frac{dh_{\mathrm{p}}}{dt} \\[2mm] \frac{dh_{\mathrm{t}}}{dt} \end{pmatrix} = \begin{pmatrix} \left(1 - \left(\frac{h_{\mathrm{p}}}{c_{\mathrm{p}}}\right)^{\alpha_1}\right) P_{\mathrm{n}}\left(1 + f_{q,1}[\boldsymbol{h}]\right) - \frac{h_{\mathrm{p}}}{c_{\mathrm{p}}}\left(2 - \frac{h_{\mathrm{p}}}{c_{\mathrm{p}}}\right) E_{\mathrm{n}}\left(1 + f_{q,2}[\boldsymbol{h}]\right) \\[2mm] 0.9\left(\frac{h_{\mathrm{p}}}{c_{\mathrm{p}}}\right)^{\alpha_1} P_{\mathrm{n}}\left(1 + f_{q,1}[\boldsymbol{h}]\right) + k_{\mathrm{exc}}\left(\frac{h_{\mathrm{t}}}{c_{\mathrm{t}}}\right)^{\alpha_3}\left(1 + f_{q,3}[\boldsymbol{h}]\right) - \frac{c_{\mathrm{t}}}{\alpha_2 - 1}\left(\frac{h_{\mathrm{t}}}{c_{\mathrm{t}}}\right)^{\alpha_2}\left(1 + f_{q,4}[\boldsymbol{h}]\right) \end{pmatrix} \\[6mm]
\textbf{(2) Flux closure law:} \\[2mm]
Q_{\mathrm{lat}} = 0.1\left(\frac{h_{\mathrm{p}}}{c_{\mathrm{p}}}\right)^{\alpha_1} P_{\mathrm{n}}\left(1 + f_{q,1}\right) + k_{\mathrm{exc}}\left(\frac{h_{\mathrm{t}}}{c_{\mathrm{t}}}\right)^{\alpha_3}\left(1 + f_{q,3}\right) + \frac{c_{\mathrm{t}}}{\alpha_2 - 1}\left(\frac{h_{\mathrm{t}}}{c_{\mathrm{t}}}\right)^{\alpha_2}\left(1 + f_{q,4}\right) \\[4mm]
\textbf{(3) Hydraulic routing PDE } \mathcal{M}_{\mathrm{hy}}\textbf{: } \partial_x Q + a_{\mathrm{kw}}\, b_{\mathrm{kw}}\, Q^{b_{\mathrm{kw}}-1} \partial_t Q = \lambda Q_{\mathrm{lat}}
\end{cases} \tag{2}$$

Now, let us explain the above governing hydrological equations. First, an interception reservoir $\mathcal{F}_{\mathrm{itc}}$ with a capacity of $c_{\mathrm{i}}$, automatically computed using the flux matching technique (Ficchì et al., 2019), allows for the computation of the neutralized rainfall $P_{\mathrm{n}}$ and neutralized evapotranspiration $E_{\mathrm{n}}$ (the neutralization of original rainfall and evapotranspiration by $\mathcal{F}_{\mathrm{itc}}$).

Then, the NN-based estimator $\phi$ which hybridizes the above differential equation system and consists of a pair of NNs, which takes as input (i) neutralized atmospheric $\mathcal{I}_{\mathrm{n}} = (P_{\mathrm{n}}, E_{\mathrm{n}})(x,t)$, along with the hydrological model states $\boldsymbol{h}(x,t)$, including production state $h_{\mathrm{p}}$ and transfer state $h_{\mathrm{t}}$, to correct spatio-temporal internal fluxes $\boldsymbol{q}(x,t)$ (process-parameterization pipeline); and (ii) physical descriptors $\boldsymbol{\mathcal{D}}(x)$ to estimate spatialized hydrological parameters $\boldsymbol{\theta}(x)$ (regionalization pipeline), as shown in Equation 3.

$$\phi : \begin{cases} \boldsymbol{f}_q(x,t) & = \phi_{\mathrm{flux}}\left(\boldsymbol{\mathcal{I}}_{\mathrm{n}}(x,t), \boldsymbol{h}(x,t); \boldsymbol{\rho}_{\mathrm{flux}}\right) \\[2mm] \boldsymbol{\theta}(x) & = \phi_{\mathrm{regio}}\left(\boldsymbol{\mathcal{D}}(x); \boldsymbol{\rho}_{\mathrm{regio}}\right) \end{cases} \tag{3}$$

with $\boldsymbol{\rho} = (\boldsymbol{\rho}_{\mathrm{flux}}, \boldsymbol{\rho}_{\mathrm{regio}})$ the vector of trainable parameters of the NN-based operator $\phi$. The vector of conceptual parameters of the chained hydrological-hydraulic model that we aim to regionalize is defined as $\boldsymbol{\theta}(x) = (\boldsymbol{\theta}_{\mathrm{rr}}(x), \boldsymbol{\theta}_{\mathrm{hy}}(x))^T$, here $\boldsymbol{\theta}_{\mathrm{rr}}(x) = (c_{\mathrm{p}}(x), c_{\mathrm{t}}(x), k_{\mathrm{exc}}(x))^T$ and $\boldsymbol{\theta}_{\mathrm{hy}}(x) = (a_{\mathrm{kw}}(x), b_{\mathrm{kw}}(x))^T$, where $c_{\mathrm{p}}$ [mm] is the capacity of the production reservoir, $c_{\mathrm{t}}$ [mm] is the capacity of the transfer reservoir, $k_{\mathrm{exc}}$ [mm/dt] is the non-conservative water exchange parameter, and $a_{\mathrm{kw}}, b_{\mathrm{kw}}$ [–] are the parameters of the kinematic wave routing. In this study, the NNs considered include either two MLPs, or an MLP for water flux correction combined with a CNN for parameter regionalization. Details on the NN architectures are provided in Appendix A.

Next, the hydrological states $\boldsymbol{h}$ are computed by solving the UDE system of Equation 2, which can be generally expressed as:

$$\frac{d\boldsymbol{h}}{dt} = \mathcal{F}_{\mathrm{gr}}\left(\,.\,, \boldsymbol{h}, \phi_{\mathrm{flux}}(\,.\,, \boldsymbol{h}; \boldsymbol{\rho})\right) \tag{4}$$

where the left-hand side represents the dynamic evolution of the system and the right-hand side is the source term that defines the dynamic, here a GR4-like operator hybridized with the following two key learnable components:

1. The NN-based operator $\phi_{\text{flux}}$ takes the hydrological model states as part of its inputs (a so-called state-dependent NN), thus affecting the model dynamics and state gradient information. It is expected to learn the model behavior by leveraging memory effects through state updates.

2. The set of physical equations $\mathcal{F}_{\text{gr}}$ with source terms integrated the NN $\phi_{\text{flux}}$ as a complementary component, which refines the internal water fluxes that describe the state dynamics. The approach allows the UDE system to preserve an original structure driven purely by physical equations, rather than directly relying on NN outputs, and enables learning under stronger physical constraints.

In this study, we consider $\mathcal{F}_{\text{gr}}$ in Equation 4 to be a set of GR production and transfer operators (UDE system of Equation 2), where $\alpha_1 = 2$ , $\alpha_2 = 5, \alpha_3 = 3.5$ are classical GR constants (cf. Perrin et al. (2003); Santos et al. (2018)); $c_{\text{p}}$, $c_{\text{t}}$, and $k_{\text{exc}}$ represent the conceptual parameters predicted by the NN $\phi_{\text{regio}}$; $f_{q,i=1..4}$ are the corrections applied to the internal fluxes, predicted by the NN $\phi_{\text{flux}}$. The bracket notation $[\boldsymbol{h}]$ in the UDE system of Equation 2 indicates that each flux correction functionally depends on $\boldsymbol{h}$, implicitly encoding the system's memory of past forcings and responses. The physical constraints are enforced by the UDE system that underlies the hydrological state-space model and can be flexibly replaced by alternative physical laws within the proposed framework.

Note that mass conservation and non-conservative exchange fluxes have been further investigated and analyzed over a large sample using an algebraic resolution of the ODE system in Huynh et al. (2025). The closure relation in Equation 2 follows a simple flux summation under the GR-like hypothesis at each pixel (for a detailed algebraic formulation, see Colleoni et al. (2025) and Huynh et al. (2025)). The numerical resolution of the hydrological UDE in Equation 2, ensuring both accurate resolution and numerical differentiability with respect to model parameters for optimization, will be explained in Section 2.2.

Finally, the routing module utilized here is based on a conceptual 1D kinematic wave model, which is numerically solved using a linearized implicit numerical scheme (Te Chow et al., 1988). Typically, the discharge routing problem is simplified to a 1D problem by adopting a "D8" drainage scheme $\mathcal{D}_\Omega(x)$, derived from processing a digital elevation model (DEM) of the terrain, with the assumption that a single pixel exhibits the largest drained area. The kinematic wave model is a PDE obtained by simplifying the 1D Saint-Venant equations, assuming the momentum is reduced to the flow friction slope, which equals the bottom slope. This is done by employing a conceptual parameterization for the momentum, $A = a_{\text{kw}} Q^{b_{\text{kw}}}$, where $A$ represents the flow's cross-sectional area, $Q$ is the discharge, and $a_{\text{kw}}$ and $b_{\text{kw}}$ are parameters to be estimated by the NN $\phi_{\text{regio}}$. This expression is inserted into the mass equation $\partial_x Q + \partial_t A = \lambda Q_{\text{lat}}$, with $Q_{\text{lat}}$ being the lateral discharge (total runoff generated at a pixel from the GR operators described above), and $\lambda$ representing the conversion factor. The result is a single-equation discharge propagation model as shown in the PDE of Equation 2. This kinematic wave model is numerically solved with a classical finite differences approach (cf. Te Chow et al. (1988)). The routing solver is implemented in a numerically differentiable form after the hydrological module, allowing derivation of a numerical adjoint for the full hydrological–routing chain and thereby enabling end-to-end gradient-based parameter optimization.

The following section will detail the numerical method used to solve the UDEs (or the standard ODE system in the absence of a state-dependent NN) in Equation 4.

## 2.2 Resolution of UDEs or ODEs within an implicit Euler scheme

To solve Equation 4, we employ an implicit Euler scheme. For a small time step $dt$, by defining $\dot{\boldsymbol{h}} = \frac{d\boldsymbol{h}}{dt}$, we have:

$$\boldsymbol{h}(t + dt) = \boldsymbol{h}(t) + dt \cdot \dot{\boldsymbol{h}}\left(\boldsymbol{h}(t + dt)\right) \tag{5}$$

Now, define:

$$\boldsymbol{g}\left(\boldsymbol{h}(t + dt)\right) = \boldsymbol{h}(t + dt) - \boldsymbol{h}(t) - dt \cdot \dot{\boldsymbol{h}}\left(\boldsymbol{h}(t + dt)\right) \tag{6}$$

Approximating the sought state $\boldsymbol{h}(t + dt)$ thus reduces to numerically solving the equation:

$$\boldsymbol{g}(y) = y - c - dt \cdot \dot{\boldsymbol{h}}(y) = \boldsymbol{0} \tag{7}$$

where $y = \boldsymbol{h}(t + dt)$ and $c = \boldsymbol{h}(t)$. Then, the solution of Equation 7 is approximated using the Newton–Raphson method as follows:

$$\begin{cases} y_0 = c, \\ y_{n+1} = y_n + \Delta y, \quad \text{where } \Delta y \text{ is the solution of } \nabla \boldsymbol{g}(y_n) \cdot \Delta y + \boldsymbol{g}(y_n) = \boldsymbol{0} \end{cases} \tag{8}$$

The Jacobian matrix $\nabla \boldsymbol{g}$ is given by:

$$\nabla \boldsymbol{g} = \begin{pmatrix} 1 - dt \frac{\partial \dot{h}_{\mathrm{p}}}{\partial h_{\mathrm{p}}} & \frac{\partial h_{\mathrm{p}}}{\partial h_{\mathrm{t}}} - dt \frac{\partial \dot{h}_{\mathrm{p}}}{\partial h_{\mathrm{t}}} \\ \frac{\partial h_{\mathrm{t}}}{\partial h_{\mathrm{p}}} - dt \frac{\partial \dot{h}_{\mathrm{t}}}{\partial h_{\mathrm{p}}} & 1 - dt \frac{\partial \dot{h}_{\mathrm{t}}}{\partial h_{\mathrm{t}}} \end{pmatrix} \approx \begin{pmatrix} 1 - dt \frac{\partial \dot{h}_{\mathrm{p}}}{\partial h_{\mathrm{p}}} & -dt \frac{\partial \dot{h}_{\mathrm{p}}}{\partial h_{\mathrm{t}}} \\ -dt \frac{\partial \dot{h}_{\mathrm{t}}}{\partial h_{\mathrm{p}}} & 1 - dt \frac{\partial \dot{h}_{\mathrm{t}}}{\partial h_{\mathrm{t}}} \end{pmatrix} \tag{9}$$

This simplification holds because $\frac{\partial h_{\mathrm{p}}}{\partial h_{\mathrm{t}}} = 0$ since production does not depend on transfer. Additionally, we assume that $h_{\mathrm{t}}$ depends on $h_{\mathrm{p}}$ only through its time derivative rather than instantaneously. Thus, $h_{\mathrm{t}}$ evolves as an accumulated effect of $h_{\mathrm{p}}$, meaning its dependence on $h_{\mathrm{p}}$ is indirect and primarily through its time derivative, justifying the approximation $\frac{\partial h_{\mathrm{t}}}{\partial h_{\mathrm{p}}} \approx 0$. The remaining terms, which are $\frac{\partial \dot{h}_{\mathrm{p}}}{\partial h_{\mathrm{p}}}$, $\frac{\partial \dot{h}_{\mathrm{p}}}{\partial h_{\mathrm{t}}}$, $\frac{\partial \dot{h}_{\mathrm{t}}}{\partial h_{\mathrm{p}}}$ and $\frac{\partial \dot{h}_{\mathrm{t}}}{\partial h_{\mathrm{t}}}$, can be derived analytically with distinct formulations depending on whether the process-parameterization NN $\phi_{\mathrm{flux}}$ is included, as follows.

First, we introduce a variable change: $\tilde{\boldsymbol{h}} = \left(\tilde{h}_{\mathrm{p}}; \tilde{h}_{\mathrm{t}}\right)^T = \left(\frac{h_{\mathrm{p}}}{c_{\mathrm{p}}}; \frac{h_{\mathrm{t}}}{c_{\mathrm{t}}}\right)^T$. Since the conceptual parameters $c_{\mathrm{p}}$ and $c_{\mathrm{t}}$ remain constant over time, their derivatives vanish, leading to: $\frac{d\tilde{h}_{\mathrm{p}}}{dt} = \frac{1}{c_{\mathrm{p}}} \frac{dh_{\mathrm{p}}}{dt}$ and $\frac{d\tilde{h}_{\mathrm{t}}}{dt} = \frac{1}{c_{\mathrm{t}}} \frac{dh_{\mathrm{t}}}{dt}$. Then, the Jacobian matrix $\nabla \boldsymbol{g}$ in Equation 9 can be computed as follows in two different cases:

1. For classical continuous state-space structure (ODE): The process-parameterization NN is absent, we set $\boldsymbol{f}_q \equiv \boldsymbol{0}$ in Equation 4. The resulting Jacobian components are:

$$\frac{\partial \dot{\tilde{h}_{\mathrm{p}}}}{\partial \tilde{h}_{\mathrm{p}}} = -\alpha_1 P_{\mathrm{n}} \tilde{h}_{\mathrm{p}}^{\alpha_1 - 1} - 2 E_{\mathrm{n}} (1 - \tilde{h}_{\mathrm{p}})$$

$$\frac{\partial \dot{\tilde{h}_{\mathrm{p}}}}{\partial \tilde{h}_{\mathrm{t}}} = 0$$

$$\frac{\partial \dot{\tilde{h}_{\mathrm{t}}}}{\partial \tilde{h}_{\mathrm{p}}} = 0.9 \alpha_1 P_{\mathrm{n}} \tilde{h}_{\mathrm{p}}^{\alpha_1 - 1}$$

$$\frac{\partial \dot{\tilde{h}_{\mathrm{t}}}}{\partial \tilde{h}_{\mathrm{t}}} = \alpha_3 k_{\mathrm{exc}} \tilde{h}_{\mathrm{t}}^{\alpha_3 - 1} - \frac{\alpha_2}{\alpha_2 - 1} c_{\mathrm{t}} \tilde{h}_{\mathrm{t}}^{\alpha_2 - 1} \tag{10}$$

   2. For hybrid continuous state-space structure (UDE): When incorporating the process-parameterization NN $\phi_{\mathrm{flux}}(.,\boldsymbol{h})$, which depends on the ODE state and predicts the model flux correction $\boldsymbol{f}_q$, one obtains a set of so-called UDEs (Equa-
tion 4). The resulting Jacobian components become:

$$\frac{\partial \dot{\tilde{h}_{\mathrm{p}}}}{\partial \tilde{h}_{\mathrm{p}}} = P_{\mathrm{n}} \left( \left(1 - \tilde{h}_{\mathrm{p}}^{\alpha_1}\right) \frac{\partial f_{q,1}}{\partial \tilde{h}_{\mathrm{p}}} - \alpha_1 \tilde{h}_{\mathrm{p}}^{\alpha_1 - 1} (1 + f_{q,1}) \right) - E_{\mathrm{n}} \left( \tilde{h}_{\mathrm{p}} \left(2 - \tilde{h}_{\mathrm{p}}\right) \frac{\partial f_{q,2}}{\partial \tilde{h}_{\mathrm{p}}} + 2 \left(1 - \tilde{h}_{\mathrm{p}}\right) (1 + f_{q,2}) \right)$$

$$\frac{\partial \dot{\tilde{h}_{\mathrm{p}}}}{\partial \tilde{h}_{\mathrm{t}}} = P_{\mathrm{n}} \left(1 - \tilde{h}_{\mathrm{p}}^{\alpha_1}\right) \frac{\partial f_{q,1}}{\partial \tilde{h}_{\mathrm{t}}} - E_{\mathrm{n}} \left(2 - \tilde{h}_{\mathrm{p}}\right) \tilde{h}_{\mathrm{p}} \frac{\partial f_{q,2}}{\partial \tilde{h}_{\mathrm{t}}}$$

$$\frac{\partial \dot{\tilde{h}_{\mathrm{t}}}}{\partial \tilde{h}_{\mathrm{p}}} = 0.9 P_{\mathrm{n}} \tilde{h}_{\mathrm{p}}^{\alpha_1 - 1} \left( \alpha_1 \left(1 + f_{q,1}\right) + \tilde{h}_{\mathrm{p}} \frac{\partial f_{q,1}}{\partial \tilde{h}_{\mathrm{p}}} \right) + k_{\mathrm{exc}} \tilde{h}_{\mathrm{t}}^{\alpha_3} \frac{\partial f_{q,3}}{\partial \tilde{h}_{\mathrm{p}}} - \frac{c_{\mathrm{t}} \tilde{h}_{\mathrm{t}}^{\alpha_2}}{\alpha_2 - 1} \frac{\partial f_{q,4}}{\partial \tilde{h}_{\mathrm{p}}}$$

$$\frac{\partial \dot{\tilde{h}_{\mathrm{t}}}}{\partial \tilde{h}_{\mathrm{t}}} = 0.9 P_{\mathrm{n}} \tilde{h}_{\mathrm{p}}^{\alpha_1} \frac{\partial f_{q,1}}{\partial \tilde{h}_{\mathrm{t}}} + k_{\mathrm{exc}} \tilde{h}_{\mathrm{t}}^{\alpha_3 - 1} \left( \alpha_3 (1 + f_{q,3}) + \tilde{h}_{\mathrm{t}} \frac{\partial f_{q,3}}{\partial \tilde{h}_{\mathrm{t}}} \right) - \frac{c_{\mathrm{t}} \tilde{h}_{\mathrm{t}}^{\alpha_2 - 1}}{\alpha_2 - 1} \left( \alpha_2 (1 + f_{q,4}) + \tilde{h}_{\mathrm{t}} \frac{\partial f_{q,4}}{\partial \tilde{h}_{\mathrm{t}}} \right) \tag{11}$$

where the NN Jacobian $\frac{\partial \boldsymbol{f}_q}{\partial \boldsymbol{h}}$ is computed explicitly using the backpropagation method, which determines the gradient of its outputs with respect to its inputs. We note that all components of the Jacobian matrix, including those from the NN, are computed directly from the analytical mathematical formulations, which significantly reduces the computational
cost compared to calling a numerical adjoint during the resolution of the ODEs with an implicit scheme. Alternative methods, which do not directly rely on mathematical formulations but could be more computationally intensive, include using linear tangents.

Finally, convergence is defined in terms of the relative Newton update measured in the Euclidean ($\ell^2$) norm:

$$\left\| \frac{\Delta \boldsymbol{h}}{\boldsymbol{h}} \right\|_2 < 10^{-6} \tag{12}$$

with a maximum of 10 Newton–Raphson iterations per time step.

The following section describes how the NNs—for both regionalization and flux correction—are trained with gradients propagated seamlessly through the entire modeling chain, including the hybrid hydrological module and the simplified hydraulic routing.

## 2.3 Model training

Considering the observed and simulated discharge time series $\boldsymbol{Q}^* = (Q^*_{g=1..N_G})^T$ and $\boldsymbol{Q} = (Q_{g=1..N_G})^T$, where $N_G$ is the number of gauges across the study domain $\Omega$, the discrepancy between the model and multi-catchment observations is evaluated using a cost function $J$ as follows:

$$J(\boldsymbol{Q}^*, \boldsymbol{Q}) = \sum_{g=1}^{N_G} w_g j(Q^*_g, Q_g) \tag{13}$$

Here, $\sum_{g=1}^{N_G} w_g = 1$ (with $w_g = 1/N_G$ in this study), and $j(Q^*_g, Q_g) = 1 - NSE(Q^*_g, Q_g)$ for each gauge. Thus, $J$ is a convex and differentiable function that relies on the output $\boldsymbol{Q}$ of the forward model $\mathcal{M}$, and consequently depends on the conceptual parameters $\boldsymbol{\theta}$ and the flux correction $\boldsymbol{f}_q$, and therefore on the parameters $\boldsymbol{\rho}$ of the NNs (cf. Equation 3). The VDA optimization problem is defined as shown in Equation 14, which involves optimizing the weights and biases of the NN-based operator $\phi$.

$$\hat{\boldsymbol{\rho}} = \arg\min_{\boldsymbol{\rho}} J\left(\boldsymbol{Q}^*, \mathcal{M}_{\mathrm{rr-hy}}\left(\,.\,; \phi(\,.\,; \boldsymbol{\rho}))\right)\right) \tag{14}$$

Note that the entire forward modeling chain—including runoff production (whether solved algebraically, with a classical ODE solver, or a UDE solver), the kinematic wave routing module, and all NN components—is fully differentiable, which enables seamless gradient backpropagation. To address the inverse problem in Equation 14, we use the Adam optimizer, a gradient-based algorithm with an adaptive learning rate suitable for non-smooth objective functions. The optimizer requires the gradient $\nabla_{\boldsymbol{\rho}} J \equiv (\nabla_{\boldsymbol{\rho}_{\mathrm{flux}}} J, \nabla_{\boldsymbol{\rho}_{\mathrm{regio}}} J)^T$ to update the weights of $\phi$. These gradients are obtained by solving the numerical adjoint model obtained by automatic differentiation with the Tapenade engine (Hascoet and Pascual, 2013), which can also be combined by chain rule to the jacobian of an external regularization NN as introduced by Huynh et al. (2024), enabling end-to-end training of the hybrid models. The NNs are trained as follows:

1. Pre-calibration of conceptual parameters: In the first step, we pre-train only the regionalization NN $\phi_{\mathrm{regio}}$ to find a spatially distributed first guess for the conceptual parameters. The weights of the process-parameterization NN $\phi_{\mathrm{flux}}$, which is embedded in the ODE solver, are initialized using a normal distribution centered at zero with a small variance. This ensures that, while the non-zero initialization allows for proper flow of gradients in the network, the outputs of $\phi_{\mathrm{flux}}$ being close to zero result in minimal corrections, thus preserving the original hydrological model structure. In other words, $\phi_{\mathrm{flux}}$ has very limited impact during this phase. The pre-training uses a relatively high learning rate of 0.004 for faster gradient descent over 40 iterations. In the case of lumped parameters (i.e., $\boldsymbol{\rho} = \bar{\boldsymbol{\theta}}$ and $\phi_{\mathrm{regio}} \equiv \mathrm{id}_{\phi_{\mathrm{regio}}}$), we simply use a heuristic optimization algorithm to provide a spatially uniform first guess.

2. Main training: In this phase, both NNs are trained simultaneously (or $\phi_{\mathrm{flux}}$ and the conceptual parameters in the case of lumped parameters). The training uses a smaller learning rate of 0.003 to ensure stability over 240 iterations. The gradients of $\phi_{\mathrm{regio}}$ are computed using a chained gradient approach described in Huynh et al. (2024), while the gradients of $\phi_{\mathrm{flux}}$, which is embedded in the differentiable Fortran code, are computed using the adjoint model.

It is worth noting that the embedded network $\phi_{\text{flux}}$ must be twice differentiable mathematically (once for solving the ODEs or UDEs, and once for the calibration process). While many activation functions in NNs are not differentiable at zero, such as ReLU (Rectified Linear Unit), stochastic optimization can often bypass this problem since the network generally produces non-zero (or non-close-to-zero) values. However, as we aim to preserve the original structure by producing outputs close to zero during pre-calibration, numerical errors can arise. To address this issue, we use the SiLU (Sigmoid Linear Unit) activation function in the hidden layers, as it is twice differentiable everywhere and provides smooth gradients.

## 3   Case study and results

### 3.1   Study area and experimental design

The models are run at a spatial resolution of 1 km and an hourly time step. The methods are evaluated using a national database covering Metropolitan France with multi-source data. The study area is the Aude River basin, located in southern France, as in Colleoni et al. (2025), but with a coarser spatial resolution of 1 km instead of 500 m. This area covers approximately 10,400 km$^2$ (corresponding to a $104 \times 100$ grid), with an active domain of 4,902 km$^2$ (4,902 active cells), and comprises 25 catchments, including 12 upstream gauges and 13 downstream gauges (Figure 1). The study period spans 9 years, from August

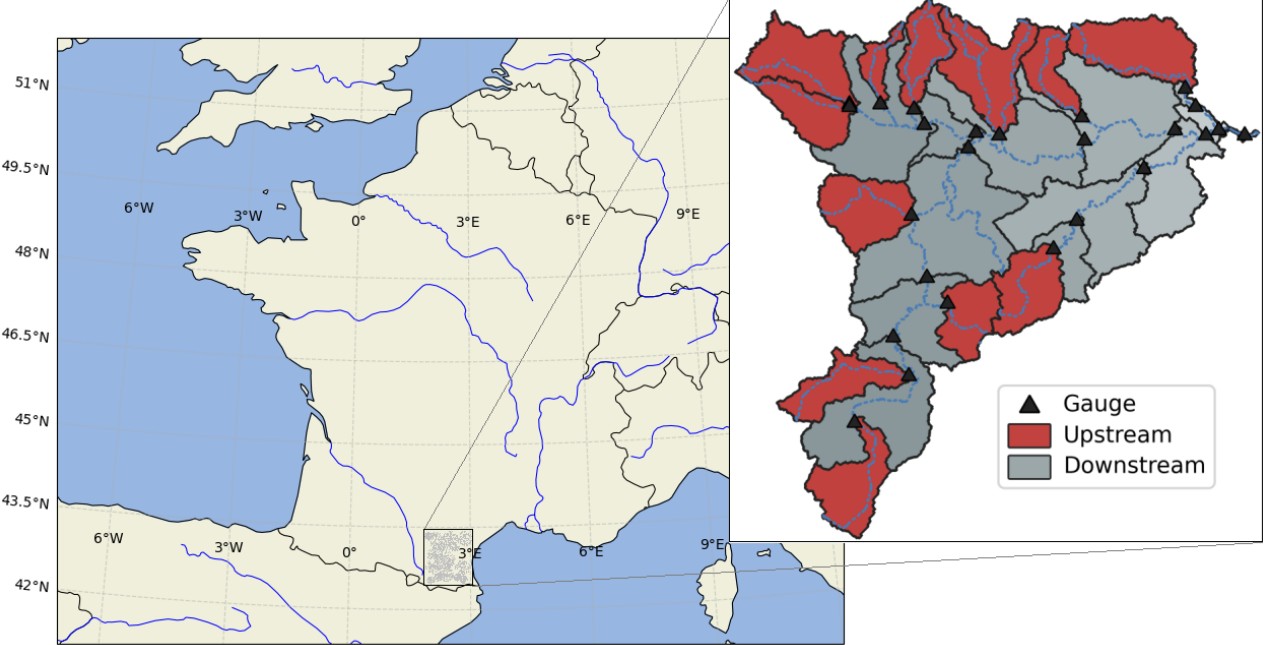

**Figure 1.** The Aude River basin, located in southern France, consists of 25 sub-catchments, including 12 upstream catchments (red-shaded regions) and 13 downstream catchments (gray-shaded regions).

2014 to July 2023, divided into two sub-periods: P1 (2015-2019) and P2 (2019-2023). We use additional one-year warm-up

periods (2014-2015 for P1 and 2018-2019 for P2) to calibrate or validate the models. Period P1 is used for calibration, while P2 is used for validation.

We perform calibration across multiple catchments using upstream gauges and evaluate regionalization performance by transferring parameters to downstream gauges, which represents a more challenging regionalization case compared to transferring parameters from downstream to upstream gauges. A set of seven descriptors with a spatial resolution of $0.01°$ in the WGS 84 projection, similar to Huynh et al. (2024), is used as inputs for the regionalization mapping $\phi_{\mathrm{regio}}$ (Figure 2).

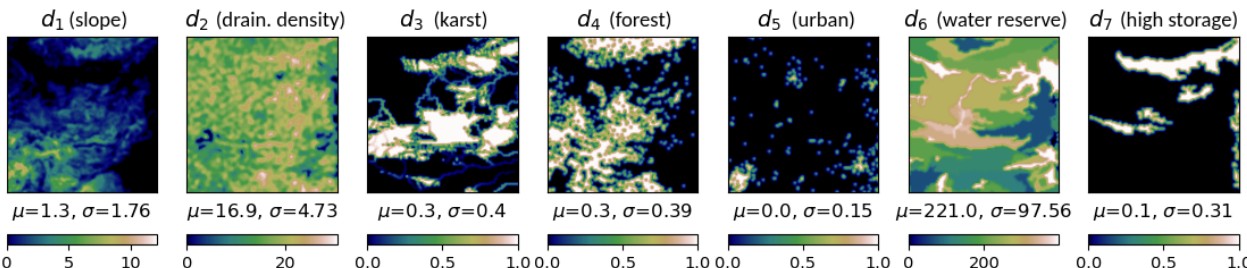

**Figure 2.** Maps of seven physical descriptors in the Aude River basin, where $\mu$ and $\sigma$ represent the spatial average and standard deviation for each descriptor. $d_1$: the local slope (in degrees); $d_2$: the drainage density; $d_3$: the percentage of basin area in karst zone; $d_4$: the forest cover rate; $d_5$: the urban cover rate; $d_6$: the potential available water reserve (in mm); and $d_7$: the high storage capacity basin rate. Before the optimization process, all descriptors are standardized between 0 and 1 using min-max scaling.

We evaluate the models based on two criteria: model structure flexibility and regionalization ability. Model structure flexibility is assessed by the model's capacity to produce interpretable water flux dynamics, both with and without the process-parameterization network $\phi_{\mathrm{flux}}$. Regionalization ability is evaluated based on the capability of the NNs $\phi_{\mathrm{regio}}$, which can be either an MLP or a CNN, to regionalize the conceptual parameters using physical descriptors. Table 1 summarizes the evaluated models and indicates the version of smash in which each was first introduced. Figure 3 illustrates the forward hydrological model through a schematic representation of the different evaluated models. The computational cost of these models is reported in Appendix B.

### 3.2 Validation of the ODE solver

A major advantage of numerically solving ODEs, as opposed to algebraic resolution methods, lies in their generalizability. While an analytical solution derived through an algebraic approach is exact, it can only be obtained under specific assumptions about the ODE system (e.g., certain fixed values of the coefficients $(\alpha_i)_{i=1..3}$ in Equation 4). In contrast, numerical methods can solve ODEs without requiring such assumptions, albeit with approximate solutions. It is thus necessary to validate the ODE solver (ODE solutions obtained by numerical scheme) against the algebraic approach before performing any numerical experiments based on this solver. This is particularly important as this work presents the first implementation of both classical and hybrid ODE solvers into a fully distributed hydrological modeling and VDA framework.

**Table 1.** Summary of evaluated models and their corresponding `smash` version. The notation consists of two parts, separated by a dot: the first part describes the model structure, while the second part indicates the mapping used to constrain the conceptual parameters.

| Notation | Model | Version | Description |
|----------|-------|---------|-------------|
| GR.U | Algebraic (lumped parameters) | v1.0 | GR4 model with lumped parameters |
| GR.MLP | Algebraic + MLP | v1.0 | Grid-based GR4 model using an MLP for regionalization |
| GR.CNN | Algebraic + CNN | v1.1 | Grid-based GR4 model using a CNN for regionalization |
| ODE.U | ODE (lumped parameters) | v1.1 | Continuous state space GR4-like model with lumped parameters |
| ODE.MLP | ODE + MLP | v1.1 | Continuous state-space GR4-like model with MLP-based regionalization |
| ODE.CNN | ODE + CNN | v1.1 | Continuous state space GR4-like model with CNN-based regionalization |
| UDE.U | UDE (lumped parameters) | v1.1 | UDE-integrated GR4-like model with lumped parameters |
| UDE.MLP | UDE + MLP | v1.1 | UDE-integrated GR4-like model with MLP-based regionalization |
| UDE.CNN | UDE + CNN | v1.1 | UDE-integrated GR4-like model with CNN-based regionalization |

To carry out this validation, we compare the GR algebraic model with lumped parameters (GR.U), which solves an analytical solution of the time-integrated ODEs, and the continuous state-space model also with lumped parameters (ODE.U), which solves the ODEs using an implicit numerical scheme. For both models, we set identical conceptual parameters and initial states, which are non-calibrated. Figure 4 shows similar simulated hydrological responses for GR.U and ODE.U obtained at the most downstream gauge. Although slight differences are found in the production state (often higher state for ODE.U) at certain periods, the transfer state for both models remains nearly identical, while streamflow simulation shows minor differences at several peak flows.

In addition to similar temporal hydrological responses, Figure 5 demonstrates nearly identical spatial patterns and values of the final model states obtained for both methods. The bias map between ODE.U and GR.U for the production state shows small deviations centered around zero, while the bias for the transfer state is nearly negligible, aligning with the earlier temporal observations. These minor discrepancies between the two approaches can be attributed primarily to the fundamental difference in their solution methodology—namely, the simultaneous numerical resolution of the ODEs for both state variables ($h_\mathrm{p}$ and $h_\mathrm{t}$) versus the exact analytical solution obtained through sequential algebraic resolution—and approximation errors of the Newton–Raphson method.

While both methods initially produce similar hydrological responses, it will be particularly interesting to observe in the next section how the model dynamics evolve during calibration, and whether differences in the numerical formulation influence model behavior and performance.

### 3.3 Model performance and interpretation

Figure 6 provides a global view of model performance across calibration and various validation scenarios. Results demonstrate that model scores in calibration increase with the complexity of regionalization mappings (from uniform to MLP and CNN). For each regionalization approach, we observe improved scores progressing from classical GR structure to ODE and UDE

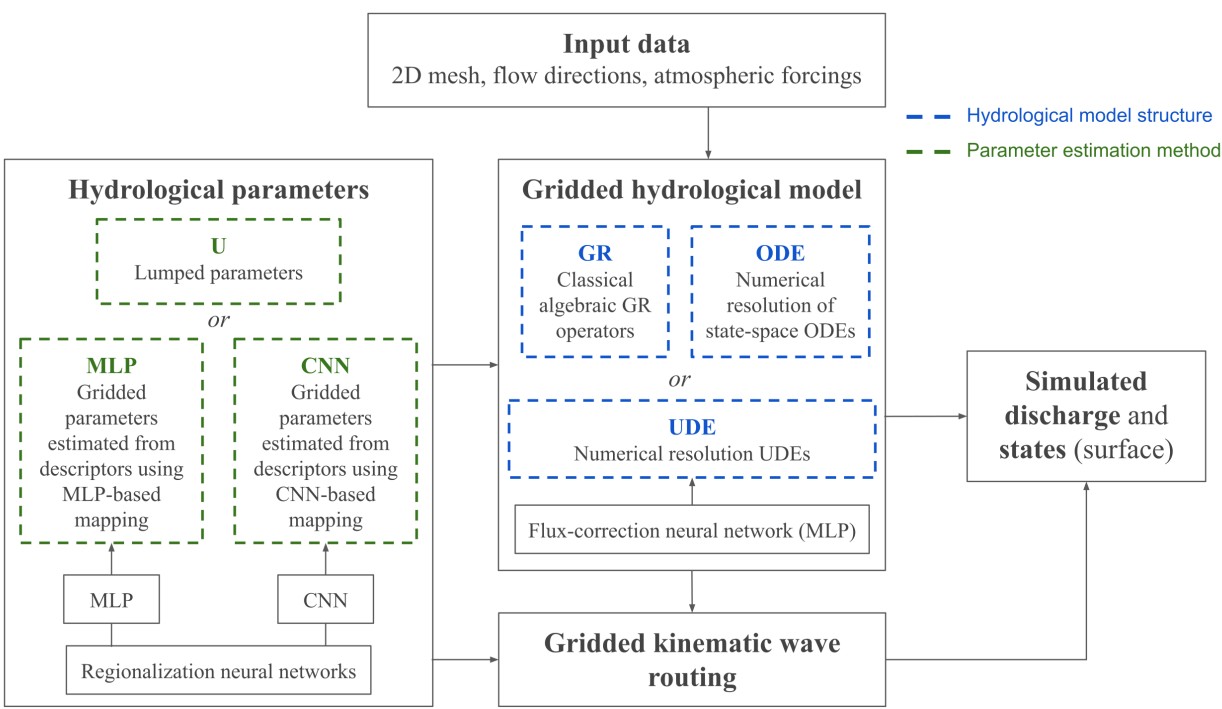

**Figure 3.** Schematic overview of the forward hydrological models. Blue blocks represent the model structures: GR denotes the GR4 structure, which uses an algebraic formulation for state updates; ODE denotes the continuous GR4-like state-space model; and UDE denotes the UDE structure with state-dependent NN for flux correction. Green blocks indicate the mappings used to constrain conceptual hydrological parameters: U represents the uniform mapping, which estimates lumped parameters without constraints; while MLP or CNN represent regionalization mappings that estimate spatially distributed hydrological parameters from physical descriptors. All models include a kinematic wave routing over a flow direction grid.

structures, exemplified by a median NSE over 0.85 for UDE.CNN compared to 0.82 for ODE.CNN and 0.8 for GR.CNN. Moreover, models with ODE and UDE structures show improved interquartile ranges and whiskers when using CNN or MLP regionalization compared to GR, while UDE.U exhibits larger variance despite maintaining higher median scores than GR.U.

For temporal validation, ODE.MLP and UDE.MLP yield similar high median scores of 0.71 compared to 0.59 for GR.MLP. Additionally, UDE.MLP demonstrates a superior interquartile range, with a 0.25-quantile of 0.65 compared to 0.57 for ODE.MLP 340 and 0.42 for GR.MLP. Models with the same regionalization approaches (uniform or CNN) show similar performance across different structures (GR, ODE, UDE), with only slight differences in median and distribution.

Spatial validation shows comparable performance across structures for each mapping, except for ODE.CNN which performs relatively worse than GR.CNN and UDE.CNN. While all models struggle with spatial validation, spatio-temporal validation yields generally good performance overall. Although ODE.CNN maintains lower performance compared to GR.CNN and 345 UDE.CNN, the MLP-based models exhibit a more stable performance across validation scenarios. For instance, ODE.MLP

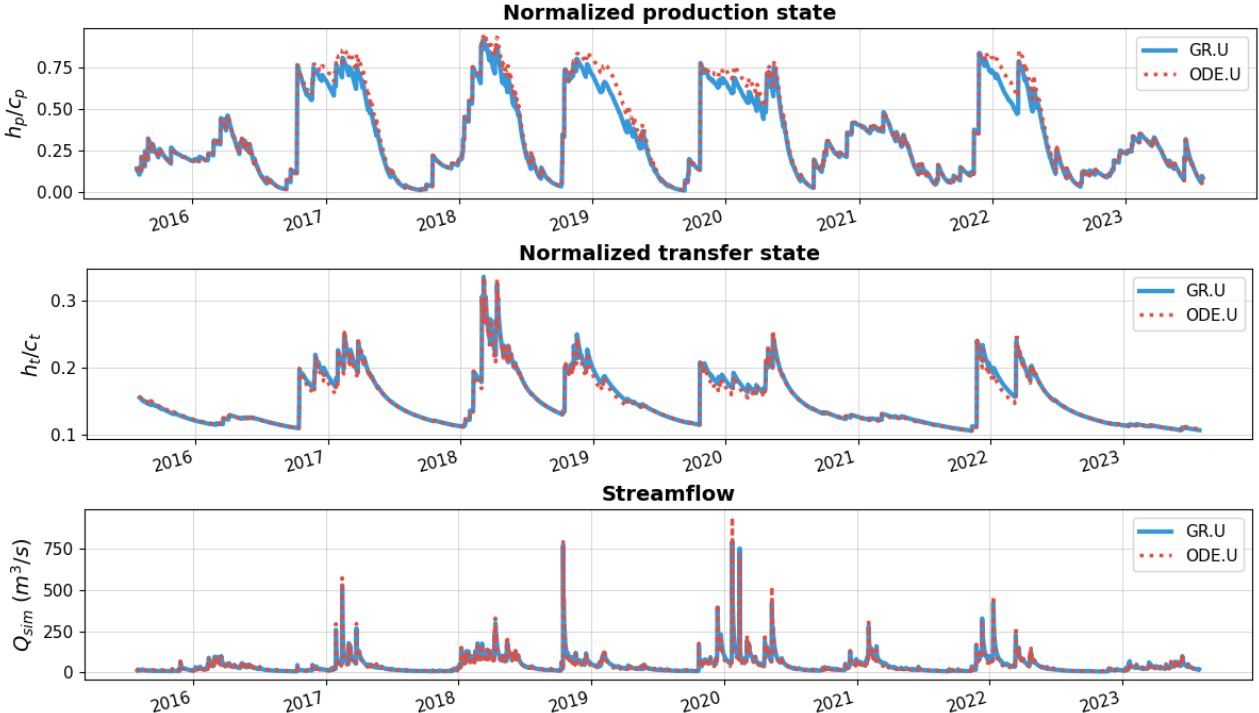

**Figure 4.** Comparison of normalized production state (top), normalized transfer state (middle), and simulated streamflow (bottom) at the most downstream gauge, obtained by algebraic resolution (GR.U) and numerical resolution (ODE.U) using the same lumped conceptual parameters and initial states.

achieves the best median scores among the three MLP-based models for spatio-temporal validation, while UDE.MLP yields the best interquartile range. Interestingly, GR.CNN exhibits promising performance in spatio-temporal validation with a median score of 0.71.

Overall, GR.CNN and UDE.MLP demonstrate consistently strong and stable performance across all calibration and vali-
350 dation scenarios. While GR.CNN generally outperforms its MLP counterpart in validation scenarios, UDE.MLP shows stable performance with good scores and favorable interquartile ranges across all validation scenarios.

To illustrate the models' flood simulation capabilities, Figure 7 presents two representative flood events during the validation period, as observed at an upstream and a downstream gauge. At both locations, all models using uniform mapping significantly underestimate the flood magnitudes, resulting in the poorest simulation performance. This result highlights the importance of regionalization approaches in accurately simulating floods, particularly in ungauged basins, where traditional methods that
use lumped parameters often fail to capture real hydrological dynamics. For the upstream gauge example, regionalization approaches using MLP and CNN produce relatively similar discharge simulations for each model structure (GR, ODE, UDE). The ODE and UDE structures with MLP show slightly better performance, with RMSE values of 4.37 and 4.13, compared to 6.2 for GR.MLP. Both CNN and MLP mappings applied to the GR structure accurately simulate the peak flow, whereas the

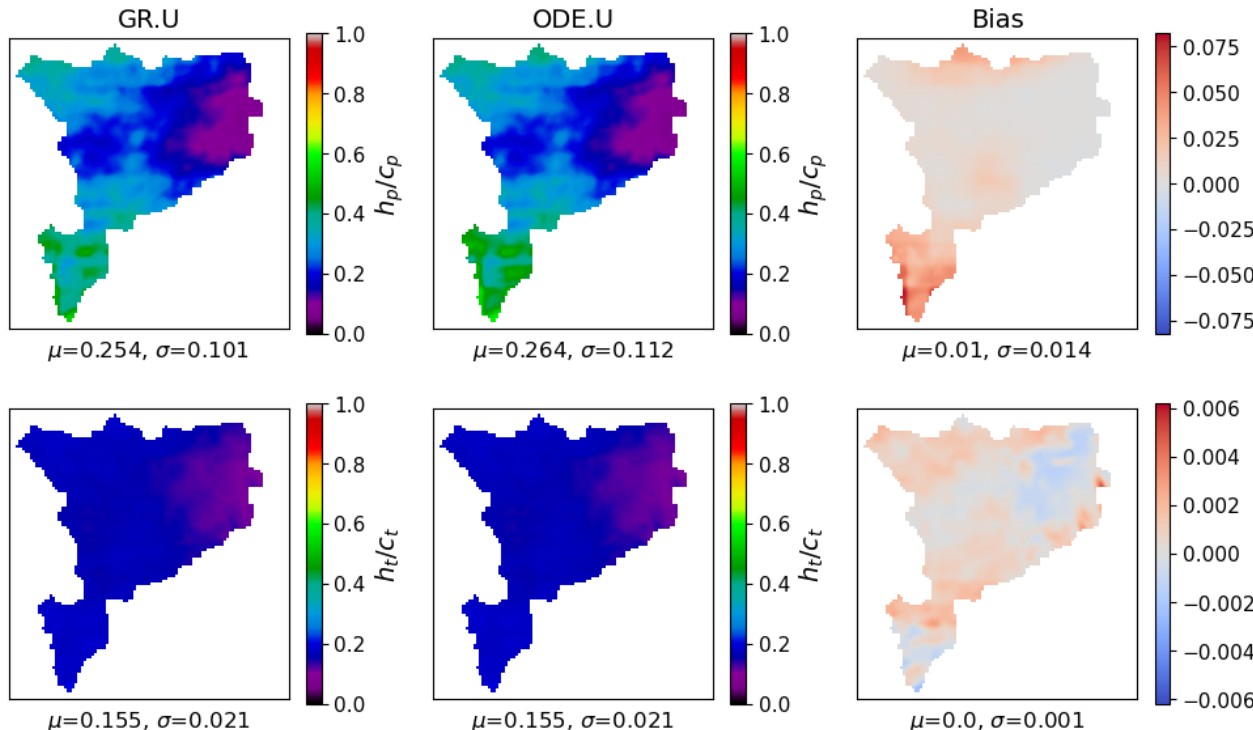

**Figure 5.** Maps of the normalized production state ($h_\mathrm{p}/c_\mathrm{p}$, top) and transfer state ($h_\mathrm{t}/c_\mathrm{t}$, bottom) at the final time step, as simulated by GR.U (left), ODE.U (middle), and the corresponding bias (right) computed as the difference between the ODE.U and GR.U states. For each map, $\mu$ and $\sigma$ denote the spatial average and standard deviation.

regionalization methods for the ODE and UDE structures tend to slightly overestimate peak flow, particularly in the case of ODE.CNN. Despite this, the timing of flood events remains accurate across all regionalization methods, with the rising limbs of the simulated hydrographs closely aligning with observations. Note that in this upstream catchment characterized by a quick hydrological response, the start of the flood event occurs nearly at the same time as the moment of heavy rainfall. In contrast, the downstream gauge event shows an approximately 8-hour delay between heavy rainfall and flood response. Most models struggle to predict the lag time correctly. This emphasizes the need to improve model realism by accounting for rainfall intensity and studying its impact in triggering non-linear flash flood responses, improving river network hydraulics, which presents a promising avenue for future research. Returning to flood simulation at the downstream, while the majority of models yield poor performance in predicting the event's beginning, generating flood responses earlier than observed, GR.CNN demonstrates impressive timing accuracy with simulations very close to observations. However, GR structure models, including GR.CNN, still underestimate flood magnitude, whereas ODE.CNN, UDE.MLP, and UDE.CNN perform more accurately.

Figure 8 shows the maps of spatially distributed conceptual parameters calibrated using CNN and MLP regionalization approaches. Overall, CNN-based models produce smoother parameter maps compared to MLP-based models, as evidenced in the

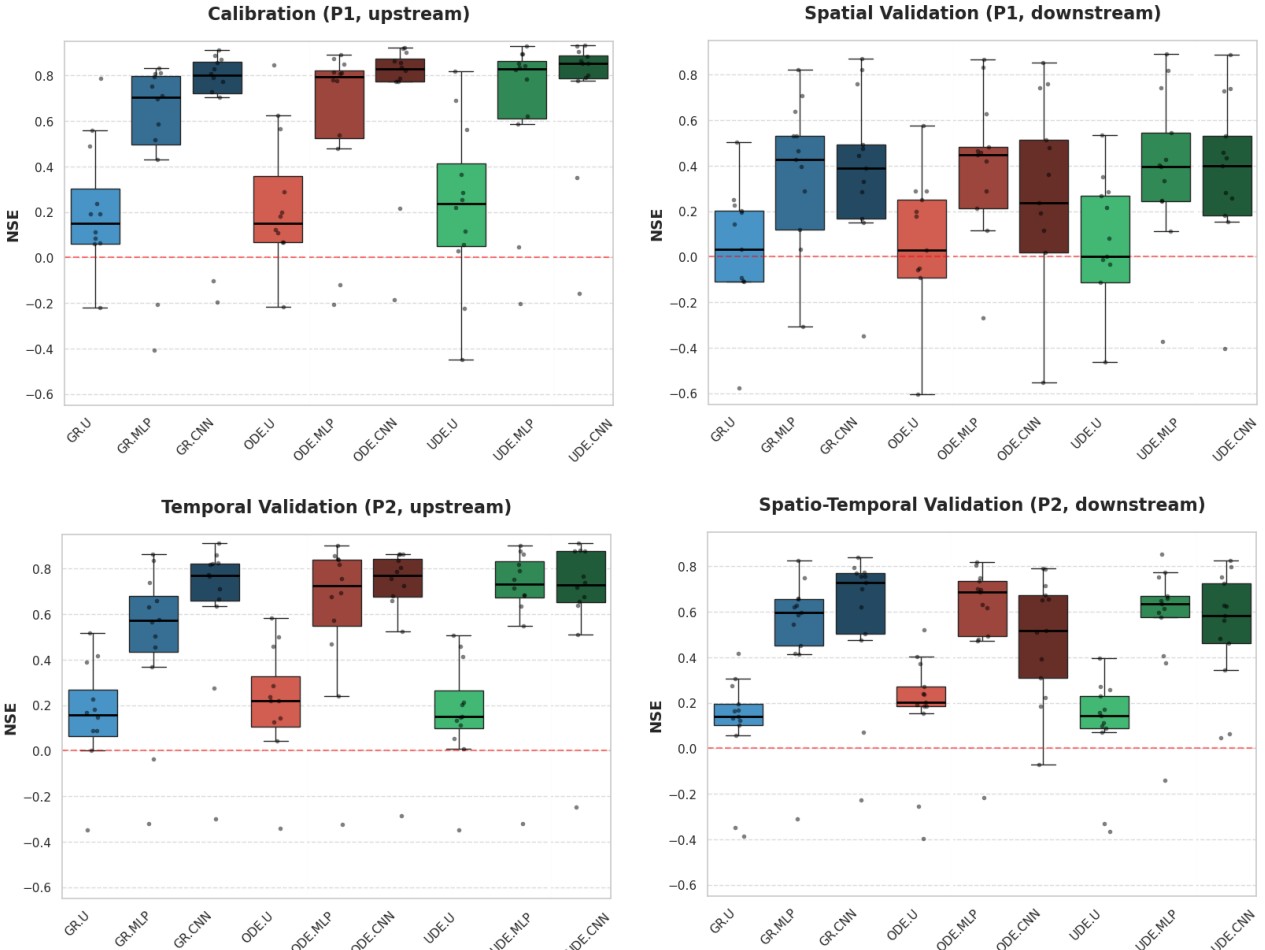

**Figure 6.** Comparison of model performance for different methods. The NSE scores are computed to evaluate: (i) calibration performance (scores computed over P1 for the 12 upstream gauges), (ii) spatial validation (scores computed over P1 for 13 downstream gauges), (iii) temporal validation (scores computed over P2 for 12 upstream gauges), and (iv) spatio-temporal validation (scores computed over P2 for downstream gauges).

maps of $c_\mathrm{p}$ and $c_\mathrm{t}$ for GR.CNN (compared to GR.MLP), $k_\mathrm{exc}$ for all CNN-based structures, $a_\mathrm{kw}$ for all CNN-based structures, and $b_\mathrm{kw}$ for GR.CNN. This smoothing effect results from the convolution operations applied to physical descriptor maps. Furthermore, we observe spatial patterns from physical descriptors (Figure 2) reflected in the parameter maps, such as the pattern at the top of the forest cover map ($d_4$) or the pattern at the bottom left of the slope ($d_1$). For each state-space structure, different regionalization mappings produce distinct parameter distributions (e.g., different patterns of GR.MLP compared to GR.CNN, ODE.MLP compared to ODE.CNN, and UDE.MLP compared to UDE.CNN). Additionally, notable differences in parameter patterns emerge across model structures, with major differences for GR-based models, while ODE and UDE structures yield

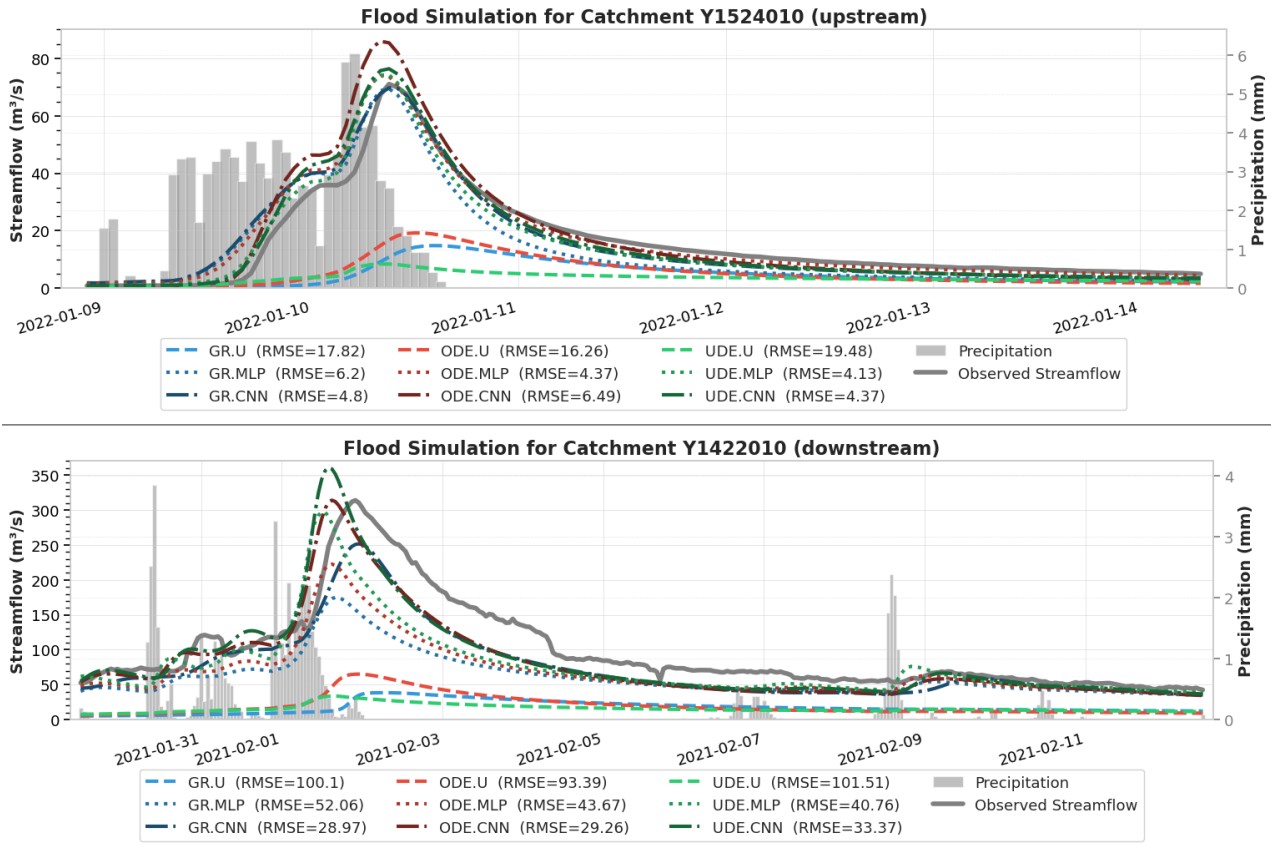

**Figure 7.** Comparison of flood simulation performance across different methods for upstream (top) and downstream (bottom) gauges.

relatively similar parameter maps (ODE.MLP compared to UDE.MLP, and ODE.CNN compared to UDE.CNN). This pattern divergence is expected since classical GR models have different state dynamics compared to continuous state-space models, even when employing the same regionalization mapping.

Figure 9 examines model dynamics through the spatial average of normalized states (production state $h_{\mathrm{p}}$ and transfer state $h_{\mathrm{t}}$). The two continuous state-space models (ODE.MLP and UDE.MLP) demonstrate similar behavior in production state, showing higher values and lower variability compared to GR.MLP. However, for the transfer state, the UDE.MLP model produces higher values and greater variability compared to the ODE.MLP and GR.MLP models. This likely results from the hybridization effect of the process-parameterization NN $\phi_{\mathrm{flux}}$, which incorporates neutralized rainfall to refine the physical equations in the ODE system. This creates a rainfall sensitivity in the transfer state, which is not accounted for in the classical GR and ODE structures.

Nevertheless, Figure 10 reveals that all three model structures exhibit relatively similar patterns in time-averaged transfer state maps, though with higher mean values (0.18 compared to 0.16 for ODE.MLP and 0.17 for GR.MLP) in the case of UDE.MLP. The maps of time-averaged production state for UDE.MLP and ODE.MLP display different patterns from those of

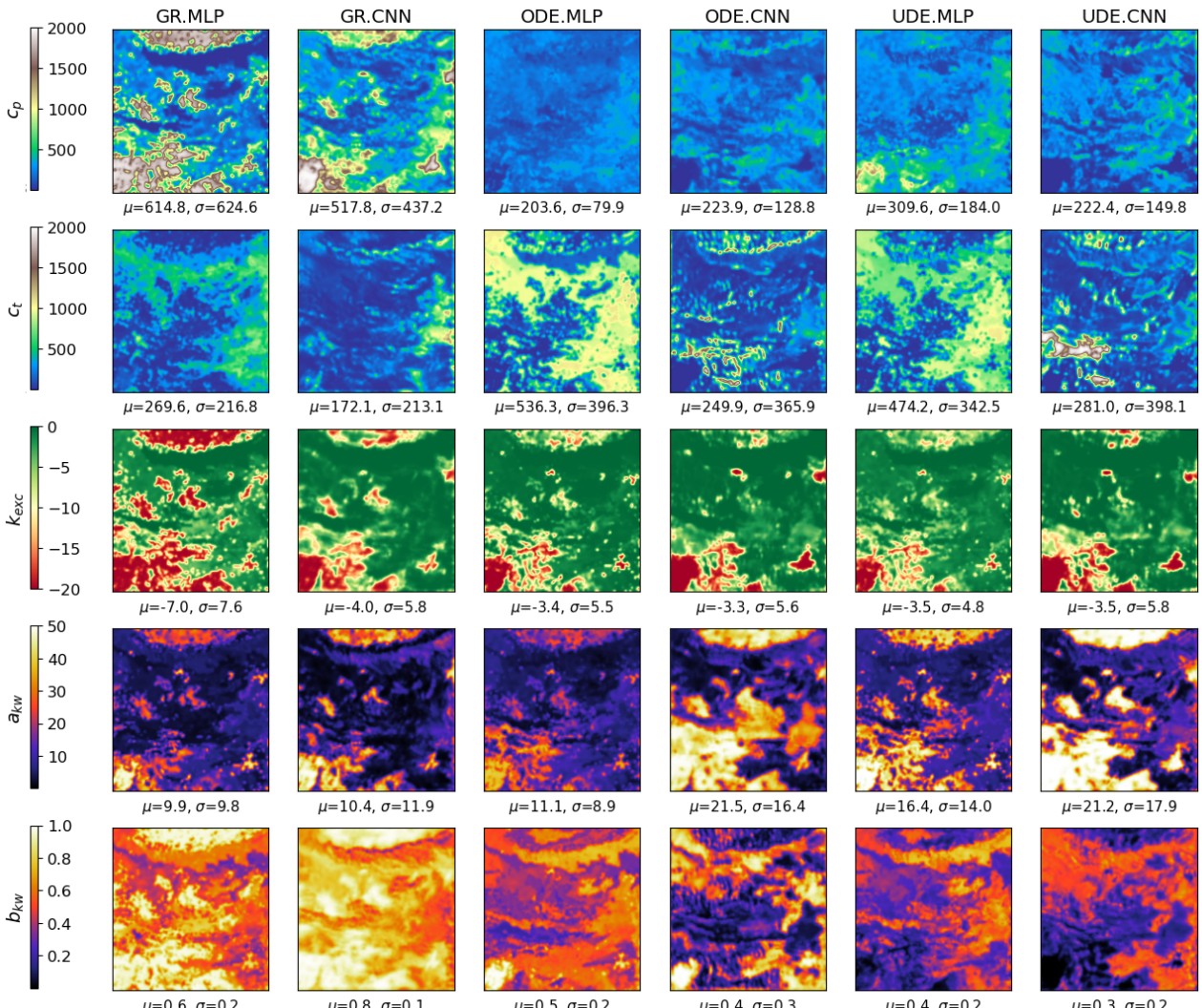

**Figure 8.** Estimated conceptual parameters across different regionalization methods. The calibrated parameters are $\boldsymbol{\theta}(x) = (c_\mathrm{p}(x), c_\mathrm{t}(x), k_\mathrm{exc}(x), a_\mathrm{kw}(x), b_\mathrm{kw}(x))^T$ with $\mu$ and $\sigma$ denoting their spatial average and standard deviation.

GR.MLP. However, no evident differences emerge between ODE.MLP and UDE.MLP for both $h_\mathrm{p}$ and $h_\mathrm{t}$. This is unsurprising as spatial hybridization effects are not expected since the process-parameterization NN employs a simple MLP (MLP of $\phi_\mathrm{flux}$ and not MLP of the regionalization mapping $\phi_\mathrm{regio}$) that does not account for or accounts less for spatial information. Future work could employ other types of $\phi_\mathrm{flux}$ to explore this aspect.

Finally, Figure 11 illustrates the hybridization effect on runoff flow (lateral discharge for routing operator) by examining high-pass filtered lateral discharge from the most downstream gauge during the validation period. A 3-day cutoff frequency high-pass filter removes seasonal and long-term patterns to focus on flood event behavior. During major events, GR.MLP

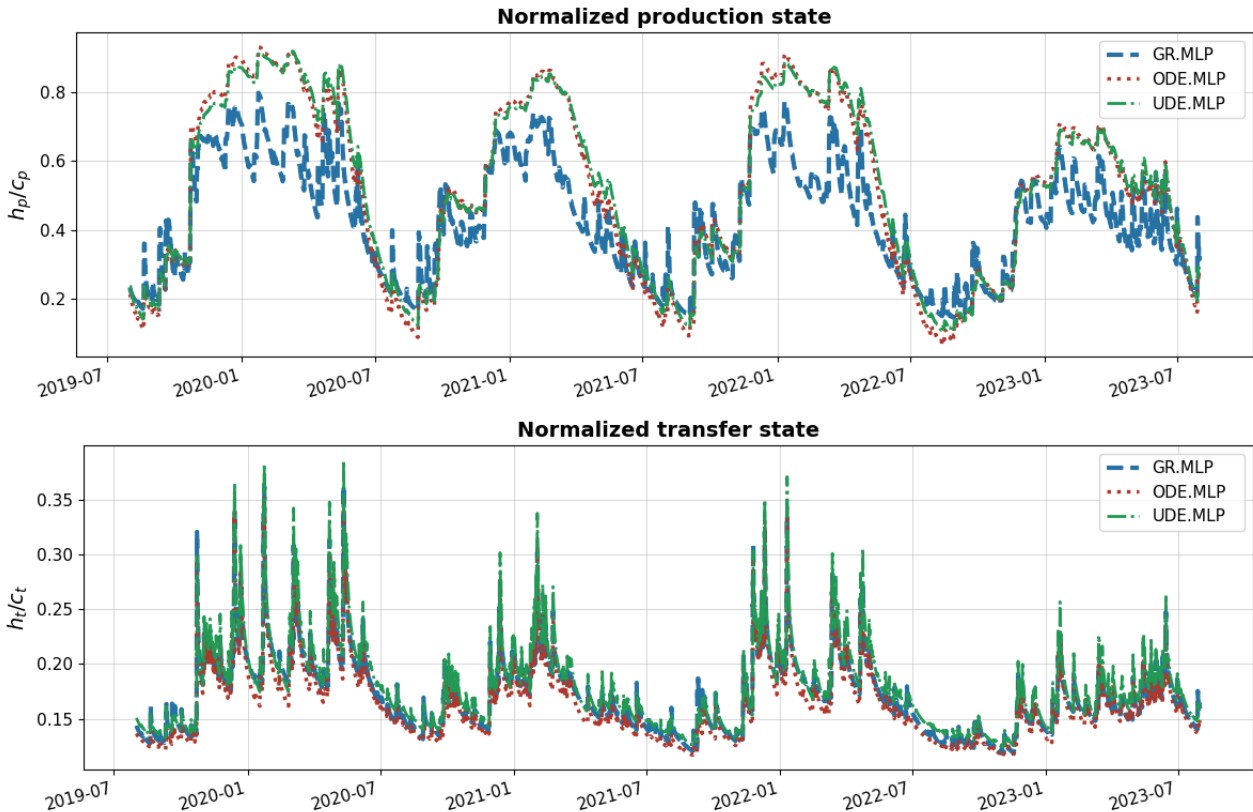

**Figure 9.** Comparison of the spatial average of the normalized production state $\overline{h_{\mathrm{p}}(t)/c_{\mathrm{p}}(t)}^{x}$ and transfer state $\overline{h_{\mathrm{t}}(t)/c_{\mathrm{t}}(t)}^{x}$ during the validation period P2, across the three model structures (GR, ODE, UDE), each employing an MLP-based regionalization mapping.

consistently produces lower lateral discharge compared to ODE.MLP. In certain situations, this results in an overestimation of flood magnitude with the ODE structure and an underestimation with the GR structure. The UDE structure effectively addresses this issue by refining internal water fluxes in the ODE system, resulting in moderate lateral discharge values that fall between those of ODE.MLP and GR.MLP. This effect is consistent with previously observed flood simulation results, demonstrating the improved hydrological response representation achieved through UDE integration.

## 4    General discussion on pure AI and hybrid modeling approaches

It has been nearly 80 years since Alan Turing first introduced the concept of a Turing machine, paving the way for the realization of thinking machines (Turing, 1950). Since then, scientists have made impressive efforts to simulate biological brain functions based on mathematical principles and the understanding of natural learning processes. AI models, with their generalization ability to learn multi-level abstractions from large datasets through backpropagation algorithms, have dramatically helped 410    automate various tasks in scientific and engineering applications (LeCun et al., 2015).

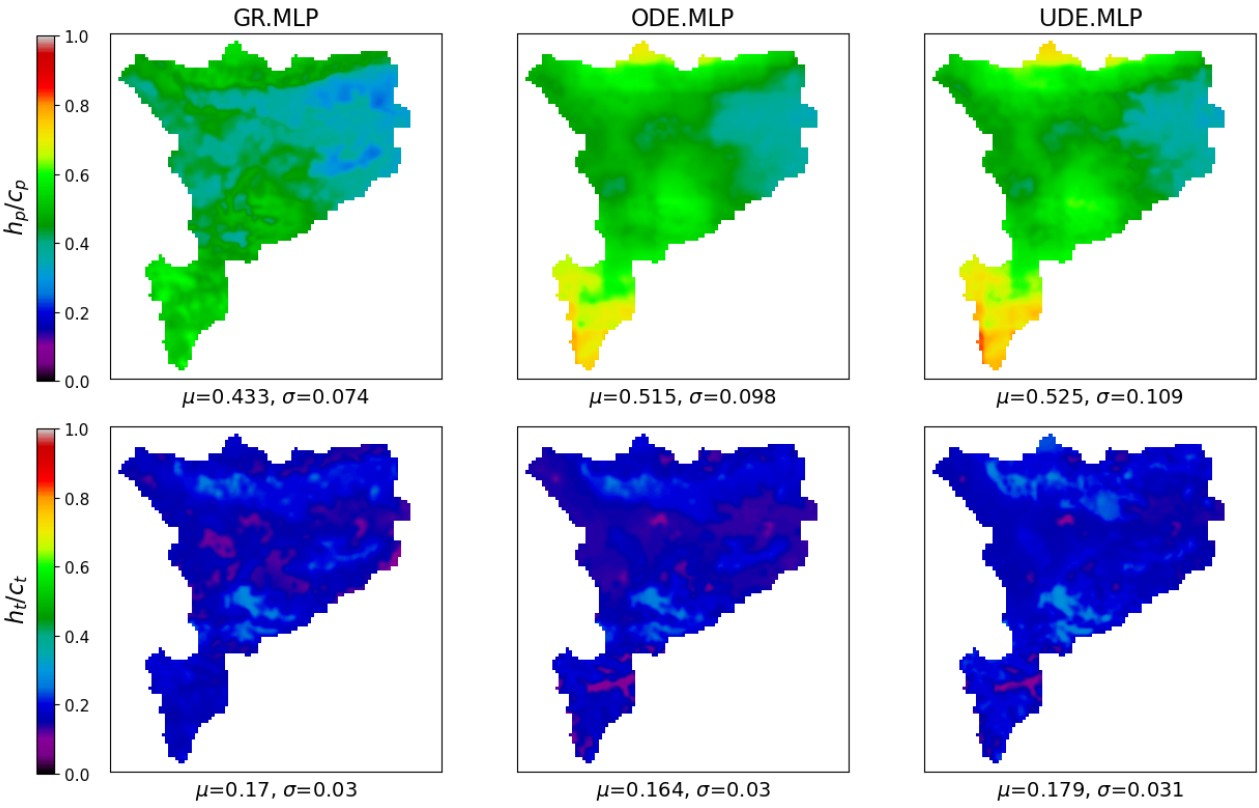

**Figure 10.** Maps of the time-averaged normalized production state $\overline{h_{\mathrm{p}}(x)/c_{\mathrm{p}}(x)}^{\,t}$ and transfer state $\overline{h_{\mathrm{t}}(x)/c_{\mathrm{t}}(x)}^{\,t}$ over the validation period P2, for the three model structures (GR, ODE, UDE), each employing an MLP-based regionalization mapping. For each map, $\mu$ and $\sigma$ denote the spatial average and standard deviation.

By 2025, AI has become ubiquitous, often perceived as a "magical" tool capable of addressing numerous challenges. Although the results of its applications seem remarkable, the evolution of AI is deeply rooted in the advancement of computational power and the application of well-established mathematical principles. The foundational concepts of modern AI are grounded in linear algebra, probability (e.g., Bayesian inference), and control theory (Lions, 1971) (see Goodfellow et al. (2016) for fundamental concepts of modern AI). Many of the key algorithms and model architectures that form the core of modern AI were developed much earlier in the 20th century but did not achieve the widespread success we see today. The primary reason for this was the lack of high-quality, extensive data and sufficient computational power to train these models effectively at that time (LeCun et al., 2015).

The intelligence we commonly reference in modern AI systems is, indeed, primarily the ability to learn patterns provided by humans through data. In other words, current AI models do not possess the capability to explore patterns beyond the given data, or even when they do, such explorations are not considered significant discoveries if researchers cannot interpret

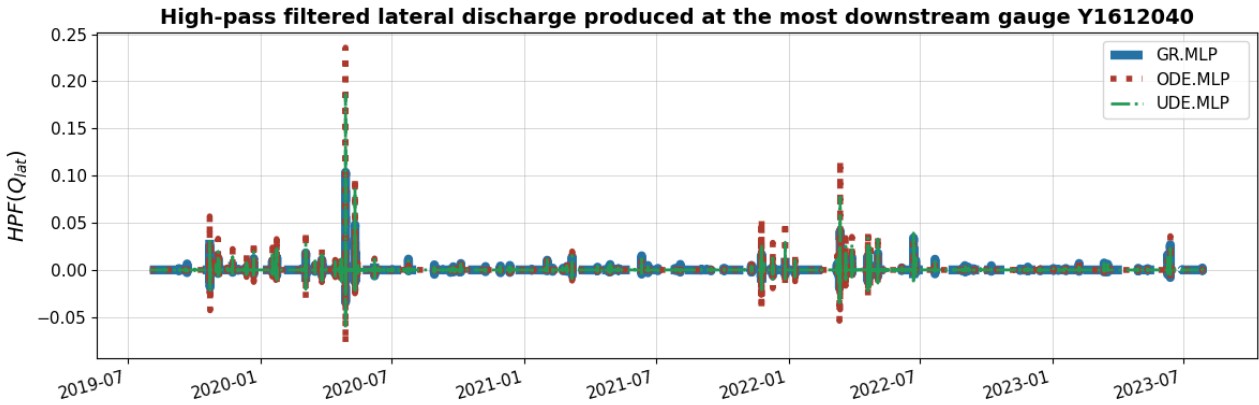

**Figure 11.** High-pass filtered lateral discharge from the most downstream gauge, using a 3-day cutoff frequency, to emphasize flood events during the validation period P2. The results are shown for 3 model structures (GR, ODE, UDE), each using MLP for regionalization.

them (LeCun, 2018). Physics, on the other hand, focuses on decoding phenomena using systems of mathematical equations, discovered through human intelligence and continually validated in pursuit of a unified theory of the universe.

The question then arises: should we focus on employing complex AI models to partially or fully replace process-based models, or should we prioritize understanding the physical interpretability of hybrid approaches, starting with simpler AI architectures? There is no straightforward answer to this question. However, based on our knowledge of both disciplines, current AI models are not capable of fully replacing process-based models, given the inherent complexity of hydrological processes and uncertainties in modeling from limited observations. Full replacement might become feasible if at least the following two conditions are met:

1. AI models become more intelligent, producing more interpretable results and demonstrating a comprehensive understanding of physics. However, this is not currently the case, as highlighted by LeCun (2022);

2. We obtain nearly perfect and complete hydrological data, accounting for all uncertainties. This requires significant effort and remains a distant goal (Beven, 2019).

Given current advancements in both hydrological modeling and AI, we believe that a comprehensive understanding of complex hydrological processes through a stand-alone AI model is not yet feasible. Even as datasets become more extensive, they remain insufficient to fully capture all relevant physical processes due to observational limitations, biases, and the inherent complexity of hydrological systems (Beven, 2019). Researchers can easily fall into the trap of optimizing for impressive performance metrics on local datasets while failing to develop models that accurately represent underlying physical mechanisms (Reichstein et al., 2019). Instead, we should focus on hybrid approaches that seamlessly integrate AI models into hydrological processes to leverage AI capabilities while maintaining physical consistency and process understanding.

## 5    Conclusions

This study proposed a differentiable and learnable framework for integrating UDEs into a process-based hydrological model, followed by a kinematic wave routing. The UDE system is solved by an implicit numerical scheme, incorporates a state-dependent NN (here is an MLP) that refines internal water fluxes and is governed by physical equations to describe the model's state dynamics. This design introduces a recurrent temporal dependency, analogous to that of RNNs, and could be extended toward an explicit recurrent architecture (e.g., RNNs, LSTMs) in future work to further enhance the model's ability to represent long-term dependencies and complex temporal interactions through additional trainable parameters. Furthermore, the regionalization mapping—used to learn the transfer function between physical descriptors and spatially distributed conceptual parameters, as introduced in Huynh et al. (2024)—is now enhanced by employing a CNN in addition to an MLP. The use of the CNN advantageously captures spatial dependencies among descriptors across the catchment and preserves the spatial structure of the parameter fields; in principle, padding strategies could be employed to up- or down-scale hydrological parameters (i.e., from coarse descriptor resolution to finer parameter resolution, or vice versa) for further study.

The proposed methods were tested on multiple catchments in the Aude River basin, demonstrating that increased complexity in model structures (from classical GR to ODE and UDE structures) and regionalization mapping (from uniform to MLP and CNN) leads to improved model calibration scores. Furthermore, models using CNN-based regionalization mapping generally exhibited smoother estimated parameter maps. This smoothness is a result of the convolution operation, which applies a relatively fine filter to the input descriptors. Evaluation scores from various validation scenarios suggest that the classical GR model using CNN (GR.CNN) and the UDE structure using MLP (UDE.MLP) demonstrated consistently strong and stable performance compared to other models. Analysis of state dynamics and runoff flow revealed the hybridization effect of the UDE structure, which modifies internal water fluxes to achieve more reliable streamflow simulations during flood events. This preliminary analysis examines recently proposed hybrid solvers for spatially distributed modeling, which could be further explored under a broader range of hydrological conditions, experimental hypotheses, and state-parameter analyses. Furthermore, the proposed UDE framework is directly applicable to other physical laws of the state-space model to infer flux correction, particularly in cases where analytic solutions of the ODEs are difficult to obtain.

Beyond the current evaluation, future work will include explicit benchmarking of the proposed hybrid physics–AI methods against classical conceptual hydrological models as well as pure DL models, such as LSTM-based rainfall–runoff models (e.g., Kratzert et al., 2018). Such comparisons will help quantify the relative benefits of embedding physical constraints versus fully data-driven learning. In addition, future analyses will explore water budget assessments using satellite-derived products to evaluate the realism of the learned flux corrections. By leveraging AI capabilities within physically-based frameworks, we aim to develop more robust, interpretable, and generalizable models for hydrological processes. This approach not only enhances the accuracy of streamflow predictions but also provides deeper insights into the underlying physical mechanisms. Additionally, it employs AI capabilities to handle massive data within a fully distributed approach for flood modeling, thereby representing a significant advancement toward a robust hybrid AI approach that emphasizes physical understanding and interpretability.

*Code and data availability.* The source code of `smash`, Version 1.1, is available and preserved on multiple platforms: GitHub at https://github.com/DassHydro/smash/tree/v1.1.0 (last access: 5 November 2025), PyPI at https://pypi.org/project/hydro-smash/1.1.0 (last access: 5 November 2025), and Zenodo with the DOI: 10.5281/zenodo.15498851. `smash` is released under the GPL-3 license and developed openly at https://github.com/DassHydro/smash (last access: 5 November 2025). The documentation is accessible at https://smash.recover.inrae.fr (last access: 5 November 2025). The dataset supporting this study comprises preprocessed data sourced from SCHAPI-DGPR and Météo-France, and are available on Zenodo with the DOI: 10.5281/zenodo.15315600. The output result files and the scripts to perform numerical experiments are available on Zenodo with the DOI: 10.5281/zenodo.16419642.

## Appendix A: Neural network architectures

The hybrid hydrological model $\mathcal{M}$ (Equation 2) embeds an NN-based operator $\phi$ (Equation 3) consisting of two components: (i) a regionalization network $\phi_{\text{regio}}$ estimating spatially distributed hydrological parameters $\boldsymbol{\theta}(x)$, and (ii) a process-parameterization network $\phi_{\text{flux}}$ correcting internal fluxes $\boldsymbol{q}(x,t)$.

### A1 Regionalization neural network $\phi_{\textbf{regio}}$

Two alternative architectures were used for the parameter regionalization pipeline. Both take as input a set of physical descriptors $\boldsymbol{\mathcal{D}}(x)$ (see Figure 2) and return the hydrological parameters $\boldsymbol{\theta}(x) = (c_{\text{p}}(x), c_{\text{t}}(x), k_{\text{exc}}(x), a_{\text{kw}}(x), b_{\text{kw}}(x))^T$. The output layer is followed by a min-max scale layer, which maps the TanH activations in $]-1,1[$ to the physically feasible ranges of the hydrological parameters, as defined in Table A1.

**Table A1.** Physical ranges and units of conceptual hydrological parameters.

| Parameter | Description | Range | Units |
|:---:|:---:|:---:|:---:|
| $c_{\text{p}}$ | Production reservoir capacity | $[1, 2000]$ | mm |
| $c_{\text{t}}$ | Transfer reservoir capacity | $[1, 2000]$ | mm |
| $k_{\text{exc}}$ | Exchange parameter | $[-20, 0]$ | mm/dt |
| $a_{\text{kw}}$ | Kinematic wave parameter | $[0.001, 50]$ | – |
| $b_{\text{kw}}$ | Kinematic wave parameter | $[0.001, 1]$ | – |

### A1.1 MLP architecture

The MLP receives the 7 physical descriptors for each individual spatial location (pixel) as a flat vector. It consists of two hidden dense layers (28 and 60 units) with ReLU activations, followed by a final dense layer of size 5 with TanH activation and scaling to parameter bounds. The architecture is summarized in Table A2. All dense layers use Glorot uniform initialization.

**Table A2.** Architecture of the MLP for $\phi_{\text{regio}}$.

| Layer | Input $\to$ Output Shape | Activation | Parameters |
|---|---|---|---|
| Dense | $7 \to 28$ | ReLU | $224$ |
| Dense | $28 \to 60$ | ReLU | $1,740$ |
| Dense | $60 \to 5$ | TanH | $305$ |
| Scale | $5 \to 5$ | – | $0$ |
| **Total** | – | – | $2,269$ |

### A1.2 CNN architecture

The CNN uses as input a spatial grid of descriptors of shape $(104, 100, 7)$ (i.e., 7 gridded physical descriptors defined on a $104 \times 100$ spatial grid). The architecture consists of a single 2D convolutional layer with 28 filters of size $4 \times 4$ and ReLU activation, followed by flattening and two dense layers of sizes 60 and 5, respectively. The last layer uses a TanH activation followed by scaling. Details are given in Table A3. All dense and convolutional layers use Glorot uniform initialization. The

**Table A3.** Architecture of the CNN for $\phi_{\text{regio}}$.

| Layer | Input $\to$ Output Shape | Activation | Parameters |
|---|---|---|---|
| Conv2D | $(104, 100, 7) \to (104, 100, 28)$ | ReLU | $3,164$ |
| Flatten | $(104, 100, 28) \to (10400, 28)$ | – | $0$ |
| Dense | $(10400, 28) \to (10400, 60)$ | ReLU | $1,740$ |
| Dense | $(10400, 60) \to (10400, 5)$ | TanH | $305$ |
| Scale and Reshape | $(10400, 5) \to (104, 100, 5)$ | – | $0$ |
| **Total** | – | – | $5,209$ |

convolutional layer employs "same" padding so that the output preserves the spatial resolution of the input descriptors, allowing
the network to produce hydrological parameters at the same grid resolution. While alternative padding strategies could be used to up- or down-scale hydrological parameters, such extensions were not investigated in this study.

### A2 Process-parameterization neural network for flux correction $\phi_{\text{flux}}$

The flux-correction network $\phi_{\text{flux}}$ is a compact MLP that takes as input 4 variables: the neutralized forcings $P_n(x,t), E_n(x,t)$ and the two state variables $h_{\text{p}}(x,t), h_{\text{t}}(x,t)$. It outputs four corrections $\boldsymbol{f}_q(x,t) = (f_{q,i}(x,t))_{i=1..4}^T$ for internal water fluxes
$\boldsymbol{q}(x,t)$. The network consists of one hidden layer of 16 neurons with SiLU activation and a final dense output layer with TanH activation (Table A4). The dense layers use He uniform initialization followed by small random scaling to limit the impact of $\phi_{\text{flux}}$ in the pre-training phase.

It is important to note that the values of $f_{q,i=1..4}$ are constrained within the range of -1 to 1 due to the TanH (Hyperbolic Tangent) activation function in the output layer of $\phi_{\text{flux}}$. Consequently, the transformation functions applied to these internal

**Table A4.** Architecture of the MLP for $\phi_{\text{flux}}$.

| Layer | Input $\rightarrow$ Output Shape | Activation | Parameters |
|---|---|---|---|
| Dense | $4 \rightarrow 16$ | SiLU | 80 |
| Dense | $16 \rightarrow 4$ | TanH | 68 |
| **Total** | – | – | 148 |

flux corrections (e.g., $1 + f_{q,1}$, $1 + f_{q,2}$, etc.) preserve the structure of the original conceptual model when $\boldsymbol{f}_q \equiv \boldsymbol{0}$, as all transformations result in a value of 1 in that case. These terms were defined according to the specific fluxes being corrected and relevant mathematical constraints.

## Appendix B: Computational time

Table B1 compares the runtime performance for all model variants. Note that the NNs and numerical schemes in the ODE-

**Table B1.** Comparison of runtime performance for different model variants, reporting forward pass time, number of trainable parameters, total optimization time, number of optimization iterations, and backward pass time for models trained using gradient-based optimization methods. Experiments were performed on an AMD EPYC 7643 48-Core Processor using 6 CPUs in parallel.

| | GR.U | GR.MLP | GR.CNN | ODE.U | ODE.MLP | ODE.CNN | UDE.U | UDE.MLP | UDE.CNN |
|---|---|---|---|---|---|---|---|---|---|
| Forward time | 18.1 s | 18.3 s | 18.4 s | 36.7 s | 38.2 s | 39.6 s | 125 s | 129 s | 135 s |
| Trainable parameters | 5 | $2,269$ | $5,209$ | 5 | $2,269$ | $5,209$ | 153 | $2,417$ | $5,357$ |
| Optimization time | 2.5 h | 9.5 h | 10 h | 3.9 h | 30.1 h | 30.5 h | 87.9 h | 176.8 h | 178.1 h |
| Total iterations | 10 | 240 | 240 | 8 | 240 | 240 | 140 | 280 | 280 |
| Backward time | – | 143 s | 150 s | – | 451.5 s | 457.5 s | $2,260$ s | $2,272$ s | $2,289$ s |

and UDE-based GR4-like models are currently not parallelized over the spatial grid due to technical difficulties in combining OpenMP with Tapenade. This limitation largely explains their substantially higher computational cost compared to the classical GR structures and could be addressed in future developments.

*Author contributions.* NNTH: methodology and conceptualization, main developer of `smash` v1.1, main writing, manuscript preparation and final redaction, numerical experiments, results analysis. PAG: methodology and conceptualization, co-developer of `smash` v1.1,
manuscript review and final redaction, results analysis, supervision and funding. FC: co-developer of `smash` v1.1, manuscript review. JM: conceptual discussions.

*Competing interests.* The authors declare that no competing interests are present.

*Financial support.* This research was supported by funding from the ANR MUFFINS project (MUltiscale Flood Forecasting with INnovating Solutions) through grant no. ANR-21-CE04-0021-01 and the NEPTUNE European project DG-ECO.

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
