# Peer review of "A hybrid physics–AI approach using universal differential equations with state-dependent neural networks for learnable, regionalizable, spatially distributed hydrological modeling"

_EGUsphere, 2025_

## Author Comment (AC2)

**Authors' Response to Editors/Reviewers of**

**Hybrid physics–AI and neural ODE approaches for spatially distributed hydrological modeling**

Huynh et al.
*GMD,*
* * *
ED: Editor Decision, RC: Reviewers' Comment,    AR: Authors' Response,    ☐ Manuscript Text

Dear Editors and Reviewers,
We greatly appreciate your time and effort in reviewing our manuscript.
We sincerely thank the two reviewers for their positive feedback highlighting the timeliness and innovation of our contribution, as well as for their very constructive suggestions, which will help us enhance the manuscript. We fully agree with the reviewers that several details and clarifications on different components of the proposed hybrid framework are needed, including a schematic presentation to better illustrate the compared models and improve clarity for a wider audience.
Responses to other points are provided in a detailed, point-by-point manner below.
Thank you again for your careful and constructive review.
N. N. T. Huynh and P.-A. Garambois, on behalf of the authors.

**1. Reviewer 1**

**1.1. Summary**

RC: The manuscript proposes a hybrid physics–AI framework for spatially distributed rainfall–runoff modeling. It builds on a differentiable, continuous state-space GR-type hydrological core with 1D routing based on a D8 drainage scheme, augmented by two neural components: (i) a regularization network that maps physical descriptors to spatially varying hydrological parameters, and (ii) a flux-correction network that adjusts internal model fluxes using meteorological inputs and prior model states. Because state evolution is governed by ODEs, the flux-correction component is trained jointly with an implicit ODE solver in a neural-ODE fashion. Across case studies, the hybrid configurations deliver more accurate and stable streamflow predictions than the classical GR baseline. Overall, this is a timely and valuable contribution to robust, differentiable coupling of process-based hydrology with machine learning.

AR: Thank you for your positive and encouraging feedback on the timeliness and value of our contribution. We appreciate your recognition of the proposed hybrid physics-AI rframework and its potential impact.

**1.2. Major comments**

RC: Self-contained presentation and background: The current manuscript would benefit from a concise background on rainfall–runoff modeling (empirical/black-box, conceptual, and distributed approaches; recent ML-based advances and their limitations) to situate the contribution and clarify what is new versus prior work.

AR: Thank you for this suggestion, we will enrich the already existing background on black-box to distributed hydrological approaches, ML-based advances and limitations. This will better highlight the main novelty of our work that lies in its generalizability when integrating NNs into an implicit numerical scheme instead of an algebraic approach (Huynh et al., 2025) to resolve the ODEs, within a hybrid regionalization framework previously developed in Huynh et al. (2024).

RC: High-level system overview: Given the number of interacting components (regionalization network, fluxcorrection network, ODE state evolution, PDE/routing), please add a clear schematic that shows data flow for the different configurations listed in Table 1.

AR: Thank you very much for this comment. We will definitely include a schematic view of the models compared in Table 1 to provide a clearer representation of the methods compared.

RC: Roles of the two neural networks and learned quantities: Different parts of the manuscript appear to attribute different parameter sets to the regionalization network. Please make explicit, in one place, which parameters each network predicts or corrects, their units and ranges, and how parameter scaling/normalization is handled to avoid ill-conditioning due to heterogeneous magnitudes.

AR: Thank you for this comment. The two NNs are described in Eq. 3, but the learned quantities and their units were not explicitly detailed as you mentioned. We will include this information in one place (probably a table) in the revised manuscript, along with their ranges and the parameter scaling/normalization approach as suggested.

RC: Notation and naming: Since the two networks serve distinct purposes, consider replacing generic labels $(\phi_1, \phi_2)$ with intuitive names.

AR: Thank you for this remark. We agree and will replace the notations $(\phi_1, \phi_2)$ with more intuitive names such as $(\phi_{flux}, \phi_{reg})$.

RC: Numerical solver choices and stability: You motivate the implicit ODE solver, which in turn needs Jacobians calculations. Please compare against an explicit solver (e.g., RK methods) in terms of accuracy, stability, computational cost, and training convergence, at least on a representative subset. Also, please report typical Newton–Raphson iteration counts, convergence criteria, and any damping/line-search strategies used to ensure robustness.

AR: Thank you for this insightful comment on the method. In fact, an explicit ODE solver was initially coded and tested in smash. We did not include this solver in the experiments and results for the following reasons:

1. Initial experiments on a simple case study showed significant errors and reduced numerical stability. This finding is consistent with previous studies (e.g., Santos et al. (2018), which employed a lumped state-space GR model—very similar to our GR-like operator distributed model—and Song et al. (2024)). It also aligns with the demonstrations in Clark and Kavetski (2010), which showed that among 8 time stepping methods for 6 hydrological model structures, the implicit Euler method provides a good solution (with a fixed time step in their study and adaptive time steps in Santos' lumped GR model, as we also implement). We will clarify this point in the revised version.

2. For model comparison, we already evaluated 9 configurations. Adding explicit solvers would introduce 6 additional configurations (3 explicit ODE solvers and 3 explicit neural ODE solvers corresponding to the 3 regionalization mappings), which would make the comparison too heavy and distract from the main focus of the paper.

For these reasons, we prefer to keep the focus on the implicit solver, but we agree that other details such as Newton-Raphson iterations, convergence criteria, etc. are important and would be more clearly reported in the revised version.

RC: PDE and differentiability: If the routing/finite-difference step (Eq. 7) is part of the training graph, please clarify whether gradients are backpropagated through the routing solver in addition to the ODE solver, and outline how this is implemented.

AR: Thank you for pointing this out. The routing is chained after the production module, and this complete forward modeling chain is differentiated to obtain the adjoint model enabling to compute cost gradient with respect to high dimensional control parameters (either hydrological ones and/or neural network ones) . This

is one of the original aspects and strength of the proposed spatially distributed modeling framework, which explicitly includes spatialized routing model (here the kinematic wave mdoel which is a hydraulic-based PDE) and hybridization possibility with neural networks adapted to predicitng quantities of a spatilized dynamic model. We will detail this point in the revised manuscript.

RC: Definition of "neutralized" inputs: Please define precisely what is meant by "neutralized" inputs and where this is applied in the pipeline.

AR: Neutralized rainfall or evapotranspiration, corresponds to the neutralization of original rainfall or evapotranspiration by the interception reservoir. This terminology is specific to GR models, as referenced in Perrin et al. (2003) and Santos et al. (2018). We will add this clarification to the revised manuscript.

RC: Closure and derivations: Provide a short derivation or a clear reference for the closure relation in Eq. (6).

AR: Thank you for this comment. The closure relation in Eq. 6, which will be more clear with the foward-inverse model flowchart that will be added, follows a simple flux summation under our GR-like hypothesis at each pixel (detailed algebraic model statement in Colleoni et al. (2025) and smash documentation: `https://smash.recover.inrae.fr/math_num_documentation/forward_structure.html#hydrological-operator-mathcal-m-rr`). We will clarify this in the revised manuscript.

RC: Model capacity and regularization: Specify the architecture sizes (layers, hidden units, activations) for both networks, parameter counts, and any regularization used.

AR: Thank you for this comment. We will add all information related the NN architecture as suggested.

RC: Training strategy: During the pre-calibration phase, have you considered training the regionalization network with the original conceptual model (i.e., without the flux-correction)? Also, how crucial was the pre-calibration to the overall training performance?

AR: Yes, during the pre-calibration phase, we train only the regionalization network (i.e., without the flux-correction). This phase is important to ensure the physical consistency of the model (i.e., meaningful parameter values) before training the flux-correction network. We will clarify this point in the revised manuscript.

**2. Reviewer 2**

RC: The authors embed a small neural network inside a physically based, gridded rainfall–runoff model and solve the resulting neural ODEs using an implicit Euler/Newton–Raphson scheme. They also learn spatially varying parameters from physical descriptors via MLPs/CNNs. In the Aude basin (France), hybrid models generally calibrate better than classical GR4 variants, and the neural-ODE approach moderates extreme runoff more plausibly during floods.

This research highlights the significance of physics-based deep learning, specifically developing a neural network to estimate fluxes and localized parameters in ODEs. It is innovative enough to be relevant to this journey. In the AI for science field, physics-guided AI is becoming increasingly important because it can be more interpretable and reliable. Additionally, it can significantly enhance the performance of traditional models based solely on physics rules.

To be honest, I am not an expert in the field of river runoff, although I have some knowledge of hybrid modeling. Therefore, for readers like myself, despite an understanding of the overall methodology, I still find it challenging to fully grasp your methods. Additionally, I would find it difficult to accept that calibrating and validating your advanced model solely within limited areas of the Aude Basin is sufficient. It would be preferable to present results from a different location to demonstrate the model's generalizability, unless one-area testing is a standard procedure in runoff modeling. Consequently, I recommend a major revision prior to the acceptance of this paper. The authors should enhance the clarity of their wording, improve the

presentation of results, and include additional validation and calibration tests.

AR: Thank you very much for your positive review and constructive suggestions. We will use them to improve the clarity of our wording and presentation so that the methodology can be more easily understood by a wider audience. The presented case is sufficient to validate the numerical consistency of the algorithm and is suitable for this software-focused journal. The proposed hybrid model formulation was built on the simpler algebraic model structure (analytically integrating the ODE set) and tested over a large sample, as reported in a previous paper. This article focuses on a numerical solver adapted to hybrid ODE systems with neural networks dependent on the model statel—a challenge especially for spatially distributed model operator. While the ODE solver's response unsurprisingly closely matches that of the algebraic model, it entails greater technical challenges and necessitates the novel methodological developments presented here.

Regarding the study area (Aude river basin), we understand your concern about relying on a medium-sized basin ($\approx 5,000 \text{km}^2$) to demonstrate our approach. We would like to highlight, however, that our previous studies have already tested related hybrid methods on large datasets: for example, the hybrid regionalization method, HDA-PR, in Huynh et al. (2024) over a study zone covering one-quarter of France and in Colleoni et al. (2025) on the CONUS dataset across the US, and the hybrid flux-correction method for the algebraic GR model structure in Huynh et al. (2025) over 235 French catchments. This technical paper focuses on model development, where the main novelty lies in a solver adapted to ODE including neural networks depending on statge, in the generalizability of the numerical scheme. Therefore, we chose to illustrate the approach on the Aude basin, a medium-sized basin with a multi-catchment setup including 25 catchments, as a representative case study.

We also note that extensive benchmarking of the different hybrid configurations against large-sample datasets requires significant computational effort, especially when integrating into a coupled 1D hydrologic–hydraulic modeling chain. Such benchmarking studies, including comparisons with purely physics-based and purely ML approaches, are already planned and will be carried out in the near future.

**2.1. Major comments**

RC: As a paper on hybrid modeling, the authors should first present the general problem with a clear governing equation. Then, explicitly show what the neural networks are doing, using clear subscripts for terms predicted by NNs, such as "QNN1" and "QNN2" in this study. Next, clearly demonstrate how the NN-predicted terms are used in the governing equations. Afterwards, show the optimization process and iterations. The authors may reorganize section 2.1 to focus their system on equations (3) and (5), removing the redundant ones, and can follow examples like Brenowitz and Bretherton (2019), Yuval and O'Gorman (2020), and Yuval et al. (2021). The goal is to make this section easier for readers to understand quickly.

AR: Thank you for this comment. Our presentation flow is as follows: Sect. 2.1 introduces the general problem (forward model and governing equations, including NN components), and Sect. 2.2 details the numerical scheme used in Sect. 2.1, with the Jacobian matrix combining NN backpropagation for the Newton–Raphson method, which is one of the key novelties of this work. The flow is organized from a view of physics and hydrological modeling, but we agree that some details on the NN components are not sufficiently reported, as also noted by Reviewer 1. We will therefore enhance the clarity of the method representation in the revised manuscript by adding details on NN architectures, learned outputs, and their physical ranges and units. However, we prefer to keep the notations of the outputs as $\boldsymbol{f}_q$ and $\boldsymbol{\theta}$, since they explicitly represent physical quantities that we aim to learn.

RC: From lines 84 to 91, what are the differences between GR4 and ODE? In their first appearance, it seems like GR4 functions like the host model. But later, it is all about NODE. However, the author also replaces many processes and parameterizations in GR with neural networks, resulting in GR.MLP and GR.CNN. The authors should be more straightforward about why they introduce both GR4 and ODE simultaneously—are they trying to show that one is better? What implications does that have? The authors should also explain GR4 and ODE more from a physics perspective at the beginning, clearly stating their purposes rather than just

referencing them. It would be helpful to include a diagram to illustrate the workflow of GR.NN, NODE.NN, and their variants.

AR: Thank you for this comment. We confirm that all models evaluated in this work are fundamentally based on the hydrological operators of the classical GR4 model. The difference between models labeled "GR" and those labeled "ODE" lies in the method used to resolve the ODEs of the state update (Eq. 5). While the original GR4 model proposes an algebraic approach (an explicit solution that only exists under specific assumptions), these ODEs can also be solved more generally with a numerical scheme (Santos et al., 2018), which we denote as "ODE" in our notations. When NNs are incorporated into this solver, we obtain a neural ODE, denoted "NODE."
The notation after the dot specifies the type of regionalization mapping used to estimate model parameters, for example, "MLP" in "NODE.MLP." These explanations are already mentioned in the manuscript, but we will rewrite them for greater clarity. As also suggested by Reviewer 1, we will add an additional diagram/flowchart to illustrate the workflow and all 9 model configurations.

RC: Include at least one subsection about neural networks, such as MLPs or CNNs. Most importantly, I still do not know what the input variables are for both NNs. Even though they are not complex neural networks, please write about their basic architecture and hyperparameters. I know some information is already in section 2.3. Please refine it and make it easier to see, such as by adding a small table, rather than hiding it within lines.

AR: Yes, we totally agree with this remark, more details on NN components will be added in the revised version of the manuscript.

RC: Although this paper focuses on hybrid models, comparing them to a pure-ML baseline would be beneficial. If it requires too much work, including references to give readers a concept of the accuracy of pure-ML models in simulating river runoffs would also be valuable.

AR: Thank you for this suggestion. We would first like to emphasize that the scope of this paper is the development of a new and more generic hybrid modeling approach (enabling to solve ODE embedding neural networks depending on states), building upon approaches that we have already rigorously tested on large-sample datasets. Our primary focus here is on resolving mathematical and numerical challenges in hybrid modeling, as well as ensuring physical interpretability, rather than on performance benchmarking.
As mentioned above, a more comprehensive evaluation of model performance (including comparisons with pure ML approaches) on larger datasets has already been planned and will be carried out shortly. We agree that such comparisons are important and appreciate your suggestion, which we will take into account in the design of our upcoming benchmarking study.

RC: The authors have shown the horizontal resolution is 500m or 1km, and a time step of 1 hour. Could they also state how many grid cells are in the region for calibration? Also how much of the GPU/CPU time are used for GRs and NODEs? Will adding the neural net components significantly add to the computational burden of the host model? In Newton iterations, please show the convergence tolerances and the usual iteration steps.

AR: Thank you for this comment. We will add all of these details about the size of the study zone, computational time, convergence criterion, etc. in the revised manuscript.

RC: It is better to add multi-basin tests (at least one contrasting basin) to demonstrate the generalization capability and robustness of the NN parameters and the NODE system.

AR: Thank you for this suggestion. While we agree that testing on multiple contrasting basins can further demonstrate the generalization capability of our methods, we emphasize that the Aude basin test case is already a medium-sized, multi-catchment setup (25 catchments) representing diverse hydrological conditions. We believe that this area provides a sufficiently complex and realistic scenario to demonstrate the proposed algorithms and their methodological innovations, particularly for a software-focused journal like GMD.
As highlighted in previous responses, the primary goal of this paper is the development and rigorous testing of the hybrid modeling framework, including the implicit solver for neural ODE and NN integration, rather

than comprehensive performance benchmarking. Extensive generalization tests across multiple basins and comparisons with pure ML and hybrid approaches are planned and will be addressed in a dedicated follow-up study, which will require significant computational effort, especially when incorporating a coupled 1D hydrologic–hydraulic modeling chain.

RC: Are the physical budgets constrained? For example, water conservation. In the runoff scenario, it would be storage = rainfall – ET – runoff. So, plot the cumulative rainfall – ET – storage – runoff closure and show how the NNs affect them.

AR: Thank you for this remark. The physical constraints are ensured by the ODE system underlying the hydrological model (with reservoir storage), which could be replaced by other physical laws within the proposed framework. Mass conservation using the same model structure—except for the ODE solver, which is one of the key novelties presented here—has been investigated in detail in Huynh et al. (2025) over a large sample. We will clarify this in the revised manuscript.

**2.2. Minor comments**

AR: We greatly appreciate your detailed minor comments in addition to the major concerns mentioned above. We will address each of these minor comments with a thorough explanation in the revision. Thank you again for your careful and constructive review of our manuscript.

**References**

Clark, M.P., Kavetski, D., 2010. Ancient numerical daemons of conceptual hydrological modeling: 1. fidelity and efficiency of time stepping schemes. Water Resources Research 46. doi:.

Colleoni, F., Huynh, N.N.T., Garambois, P.A., Jay-Allemand, M., Organde, D., Renard, B., De Fournas, T., El Baz, A., Demargne, J., Javelle, P., 2025. Smash v1.0: A differentiable and regionalizable high-resolution hydrological modeling and data assimilation framework. EGUsphere 2025, 1–36. doi:.

Huynh, N.N.T., Garambois, P.A., Colleoni, F., Renard, B., Roux, H., Demargne, J., Jay-Allemand, M., Javelle, P., 2024. Learning regionalization using accurate spatial cost gradients within a differentiable high-resolution hydrological model: Application to the french mediterranean region. Water Resources Research 60, e2024WR037544. doi:.

Huynh, N.N.T., Garambois, P.A., Renard, B., Colleoni, F., Monnier, J., Roux, H., 2025. A distributed hybrid physics–ai framework for learning corrections of internal hydrological fluxes and enhancing high-resolution regionalized flood modeling. Hydrology and Earth System Sciences 29, 3589–3613. doi:.

Perrin, C., Michel, C., Andrèassian, V., 2003. Improvement of a parsimonious model for streamflow simulation. Journal of hydrology 279, 275–289. doi:.

Santos, L., Thirel, G., Perrin, C., 2018. Continuous state-space representation of a bucket-type rainfall-runoff model: a case study with the gr4 model using state-space gr4 (version 1.0). Geoscientific Model Development 11, 1591–1605. doi:.

Song, Y., Knoben, W.J.M., Clark, M.P., Feng, D., Lawson, K., Sawadekar, K., Shen, C., 2024. When ancient numerical demons meet physics-informed machine learning: adjoint-based gradients for implicit differentiable modeling. Hydrology and Earth System Sciences 28, 3051–3077. doi:.

---

## Author Response (AR1)

**Authors' Response to Editors/Reviewers of**

**A hybrid physics–AI approach using universal differential equations with state-dependent neural networks for learnable, regionalizable, spatially distributed hydrological modeling**

Huynh et al.
*GMD,*
* * *
ED: Editor Decision, RC: Reviewers' Comment,    AR: Authors' Response,    ☐ Manuscript Text

Dear Editors and Reviewers,

We greatly appreciate your time and effort in reviewing our manuscript.

We sincerely thank the two reviewers for their positive feedback highlighting the timeliness and innovation of our contribution, as well as for their very constructive suggestions that have helped us enhance the manuscript. We fully agree with the reviewers that several details and clarifications on different components of the proposed hybrid framework are needed, including a schematic presentation to better illustrate the compared models and improve clarity for a wider audience. We also note that the paper title, abstract, and introduction section have been significantly enhanced to clarify our unique contributions to the topic of hybrid hydrological modeling.

Please find the revised manuscript in the attached file. Our responses to the reviewers' comments are provided in detail, point by point, below.

Thank you again for your careful and constructive review.

N. N. T. Huynh and P.-A. Garambois, on behalf of the authors.

**1. Reviewer 1**

**1.1. Summary**

RC:    The manuscript proposes a hybrid physics–AI framework for spatially distributed rainfall–runoff modeling. It builds on a differentiable, continuous state-space GR-type hydrological core with 1D routing based on a D8 drainage scheme, augmented by two neural components: (i) a regionalization network that maps physical descriptors to spatially varying hydrological parameters, and (ii) a flux-correction network that adjusts internal model fluxes using meteorological inputs and prior model states. Because state evolution is governed by ODEs, the flux-correction component is trained jointly with an implicit ODE solver in a neural-ODE fashion. Across case studies, the hybrid configurations deliver more accurate and stable streamflow predictions than the classical GR baseline. Overall, this is a timely and valuable contribution to robust, differentiable coupling of process-based hydrology with machine learning.

AR:    Thank you for your positive and encouraging feedback on the timeliness and value of our contribution. We appreciate your recognition of the proposed hybrid physics-AI rframework and its potential impact.

**1.2. Major comments**

RC:    Self-contained presentation and background: The current manuscript would benefit from a concise background on rainfall–runoff modeling (empirical/black-box, conceptual, and distributed approaches; recent ML-based advances and their limitations) to situate the contribution and clarify what is new versus prior work.

AR:    Thank you for this suggestion. We have made significant efforts to enhance the literature review in the background section and to clarify the contribution of this work. The introduction section is now separated into 3 subsections. It begins with an overview of the evolution of rainfall–runoff modeling approaches, and

highlighting the motivation for employing data-driven methods, as well as the rise of AI and its applications in hydrology: we discuss the two main ways in which ML/DL techniques have been utilized in hydrology and outline their current limitations. Then, in the second subsection, we emphasize the need for hybrid approaches in spatially distributed modeling—a topic that has received relatively little attention—and explain why integrating NNs directly into physical models and numerical solvers offers distinct advantages and is critical for advancing distributed hydrological modeling. Finally, we have strengthened the last paragraphs of the introduction to better highlight the novelty and main contributions of our study in light of this background. The combination of the following three points makes our contributions unique: (i) the need for a hybrid approach for spatially distributed models; (ii) integration of a state-dependent NN into the ODE system for the correction of specific source terms which form UDEs with state-dependent NN, and (iii) resolution of the UDE with an implicit numerical scheme. Especially, the last two points remains unexplored in hydrological modeling, which requires the efficient computation of the Jacobian matrix for state-dependent NNs within the UDE system for its resolution, and the derivation of a numerical adjoint of the complete hydrological model including UDE and gridded kinematic wave (PDE) routing to enable high-dimensional parameter optimization. These points have been clarified in the last paragraph of subsection 1.2 and the first paragraph of subsection 1.3.

RC: High-level system overview: Given the number of interacting components (regionalization network, flux-correction network, ODE state evolution, PDE/routing), please add a clear schematic that shows data flow for the different configurations listed in Table 1.

AR: Thank you very much for this comment. A schematic view of the forward hydrological models has been added (Fig. 3) in addition to Table 1 to provide a clearer representation of the methods compared.

RC: Roles of the two neural networks and learned quantities: Different parts of the manuscript appear to attribute different parameter sets to the regionalization network. Please make explicit, in one place, which parameters each network predicts or corrects, their units and ranges, and how parameter scaling/normalization is handled to avoid ill-conditioning due to heterogeneous magnitudes.

AR: Thank you for this comment. The revised manuscript has clarified this concern, indicating the learned quantities and their units, also all details on NNs architecture, normalization/scaling methods have been added in Appendix A.

RC: Notation and naming: Since the two networks serve distinct purposes, consider replacing generic labels $(\phi_1, \phi_2)$ with intuitive names.

AR: Thank you for this remark. We agree and have replaced the notations $(\phi_1, \phi_2)$ with more intuitive names such as $(\phi_{flux}, \phi_{regio})$.

RC: Numerical solver choices and stability: You motivate the implicit ODE solver, which in turn needs Jacobians calculations. Please compare against an explicit solver (e.g., RK methods) in terms of accuracy, stability, computational cost, and training convergence, at least on a representative subset. Also, please report typical Newton–Raphson iteration counts, convergence criteria, and any damping/line-search strategies used to ensure robustness.

AR: Thank you for this insightful comment on the method. In fact, an explicit ODE solver was initially coded and tested in smash. We did not include this solver in the experiments and results for the following reasons:

1. Initial experiments on a simple case study showed significant errors and reduced numerical stability with the explicit solver applied to this GR-like hybrid model. This finding is consistent with previous studies (e.g., Santos et al. (2018), which employed a lumped state-space GR model—very similar to our GR-like operator distributed model—and Song et al. (2024)). It also aligns with the demonstrations in Clark and Kavetski (2010), which showed that among 8 time stepping methods for 6 hydrological model structures, the implicit Euler method provides a good solution (with a fixed time step in their

study and adaptive time steps in Santos' lumped GR model, as we also implement). We have clarified this point in the introduction section.

2. For model comparison, we already evaluated 9 configurations. Adding explicit solvers would introduce 6 additional configurations (3 explicit ODE solvers and 3 explicit UDE solvers corresponding to the 3 regionalization mappings), which would make the comparison too heavy and distract from the main focus of the paper.

For these reasons, we prefer to keep the focus on the implicit solver, but we agree that other details such as Newton-Raphson iterations, convergence criteria, etc. are important and have been clearly reported in Sect. 2.2 of the revised version.

RC: PDE and differentiability: If the routing/finite-difference step (Eq. 7) is part of the training graph, please clarify whether gradients are backpropagated through the routing solver in addition to the ODE solver, and outline how this is implemented.

AR: Thank you for pointing this out. The routing is chained after the production module, and this complete forward modeling chain is differentiated to obtain the adjoint model enabling to compute cost gradient with respect to high dimensional control parameters (either hydrological ones and/or neural network ones). This is one of the original aspects and strength of the proposed spatially distributed modeling framework, which explicitly includes spatialized routing model (here the kinematic wave model which is a hydraulic-based PDE) and hybridization possibility with neural networks adapted to predicting quantities of a spatialized dynamic model. We hope that the added schematic view of the forward model (Fig. 3) and the added explanation in Sect. 2.1 and 2.3 have clarified this point.

RC: Definition of "neutralized" inputs: Please define precisely what is meant by "neutralized" inputs and where this is applied in the pipeline.

AR: Neutralized rainfall or evapotranspiration, corresponds to the neutralization of original rainfall or evapotranspiration by the interception reservoir. This terminology is specific to GR models, as referenced in Perrin et al. (2003) and Santos et al. (2018). This point has been clarified in the revised manuscript.

RC: Closure and derivations: Provide a short derivation or a clear reference for the closure relation in Eq. (6).

AR: Thank you for this comment. The closure relation in Eq. 6 follows a simple flux summation under our GR-like hypothesis at each pixel (detailed algebraic model statement in Colleoni et al. (2025). We have clarified this in Eq. (2) and in Sect. 2.1 of the revised manuscript.

> The closure relation in Equation 2 follows a simple flux summation under the GR-like hypothesis at each pixel (for a detailed algebraic formulation, see Colleoni et al. (2025) and Huynh et al. (2025)).

RC: Model capacity and regularization: Specify the architecture sizes (layers, hidden units, activations) for both networks, parameter counts, and any regularization used.

AR: Thank you for this comment. A new appendix including all details related the NN architecture has been added in the revised manuscript.

RC: Training strategy: During the pre-calibration phase, have you considered training the regionalization network with the original conceptual model (i.e., without the flux-correction)? Also, how crucial was the pre-calibration to the overall training performance?

AR: Yes, during the pre-calibration phase, we train only the regionalization network (i.e., without the flux-correction). This phase is important to ensure the physical consistency of the model (i.e., meaningful parameter values) before training the flux-correction network. This has been already explained in Sect. 2.3 of the manuscript but we have also detailed in Appendix A in this revised version.

**2. Reviewer 2**

RC: The authors embed a small neural network inside a physically based, gridded rainfall–runoff model and solve the resulting neural ODEs using an implicit Euler/Newton–Raphson scheme. They also learn spatially varying parameters from physical descriptors via MLPs/CNNs. In the Aude basin (France), hybrid models generally calibrate better than classical GR4 variants, and the neural-ODE approach moderates extreme runoff more plausibly during floods.

This research highlights the significance of physics-based deep learning, specifically developing a neural network to estimate fluxes and localized parameters in ODEs. It is innovative enough to be relevant to this journey. In the AI for science field, physics-guided AI is becoming increasingly important because it can be more interpretable and reliable. Additionally, it can significantly enhance the performance of traditional models based solely on physics rules.

To be honest, I am not an expert in the field of river runoff, although I have some knowledge of hybrid modeling. Therefore, for readers like myself, despite an understanding of the overall methodology, I still find it challenging to fully grasp your methods. Additionally, I would find it difficult to accept that calibrating and validating your advanced model solely within limited areas of the Aude Basin is sufficient. It would be preferable to present results from a different location to demonstrate the model's generalizability, unless one-area testing is a standard procedure in runoff modeling. Consequently, I recommend a major revision prior to the acceptance of this paper. The authors should enhance the clarity of their wording, improve the presentation of results, and include additional validation and calibration tests.

AR: Thank you very much for your positive review and constructive suggestions. We have used them to improve the clarity of our wording and presentation so that the methodology can be more easily understood by a wider audience. We understand your concern on large sample validation of the hybrid physics-AI hydrological model presented, which as already been done in Huynh et al. (2025) on a large catchment sample for the algebraic solver version while the present research focuses on its formulation with a fully equivalent UDE and its implicit resolution. This UDE solver is clearly validated on the original algebraic one on several cases addressing a wide range of states and forcing conditions which is sufficient for validating a UDE solver against its reference algebraic version.

Regarding the study area (Aude river basin), we understand your concern about relying on a medium-sized basin ($\approx 5,000\text{km}^2$) to demonstrate our approach. We would like to highlight, however, that our previous studies have already tested related hybrid methods on large datasets: for example, the hybrid regionalization method, HDA-PR, in Huynh et al. (2024) over a study zone covering one-quarter of France and in Colleoni et al. (2025) on the CONUS dataset across the US, and the hybrid flux-correction method for the algebraic GR model structure in Huynh et al. (2025) over 235 French catchments. This research paper focuses on model development, where the main novelty lies in a solver adapted to ODE including neural networks depending on the model states, in the generalizability of the numerical scheme. Therefore, we chose to illustrate the approach on the Aude basin, a medium-sized basin with a multi-catchment setup including 25 catchments, as a representative case study.

We also note that extensive benchmarking of the different hybrid configurations against large-sample datasets requires significant computational effort, especially when integrating into a coupled 1D hydrologic–hydraulic modeling chain. Such benchmarking studies, including comparisons with purely physics-based and purely ML approaches, are already planned and will be carried out in the near future.

**2.1. Major comments**

RC: As a paper on hybrid modeling, the authors should first present the general problem with a clear governing equation. Then, explicitly show what the neural networks are doing, using clear subscripts for terms predicted by NNs, such as "QNN1" and "QNN2" in this study. Next, clearly demonstrate how the NN-predicted terms are used in the governing equations. Afterwards, show the optimization process and iterations. The authors may reorganize section 2.1 to focus their system on equations (3) and (5), removing the redundant ones, and

can follow examples like Brenowitz and Bretherton (2019), Yuval and O'Gorman (2020), and Yuval et al. (2021). The goal is to make this section easier for readers to understand quickly.

AR: Thank you for this comment. The methodology section has been fully rewritten following your advices and we believe that it is now clearer to present governing hydrological equations first and then the NN hybridization, highlighting the originality of having a grid based hydrological-hydraulic chain with regionalization and sate-dependent NNs, its numerical differentiability and parameters learning capabilities.
Our presentation flow is as follows: Sect. 2.1 introduces the general problem (forward model and governing equations, including NN components), and Sect. 2.2 details the numerical scheme used in Sect. 2.1, with the Jacobian matrix combining NN backpropagation for the Newton–Raphson method, which is one of the key novelties of this work. The flow is organized from a view of physics and hydrological modeling, but we agree that some details on the NN components are not sufficiently reported, as also noted by Reviewer 1. Thus, we have made significant efforts to enhance the clarity of the method representation in the revised manuscript by adding details on NN architectures, learned outputs, their physical ranges/units (Appendix A), and schematic view of the evaluated hydrological model (Fig. 3). However, we prefer to keep the notations of the outputs as $\boldsymbol{f}_q$ and $\boldsymbol{\theta}$, since they explicitly represent physical quantities that we aim to learn.

RC: From lines 84 to 91, what are the differences between GR4 and ODE? In their first appearance, it seems like GR4 functions like the host model. But later, it is all about NODE. However, the author also replaces many processes and parameterizations in GR with neural networks, resulting in GR.MLP and GR.CNN. The authors should be more straightforward about why they introduce both GR4 and ODE simultaneously—are they trying to show that one is better? What implications does that have? The authors should also explain GR4 and ODE more from a physics perspective at the beginning, clearly stating their purposes rather than just referencing them. It would be helpful to include a diagram to illustrate the workflow of GR.NN, NODE.NN, and their variants.

AR: Thank you for this comment. We confirm that all models evaluated in this work are fundamentally based on the hydrological operators of the classical GR4 model. The difference between models labeled "GR" and those labeled "ODE" lies in the method used to resolve the ODEs of the state update (Eq. 5). While the original GR4 model proposes an algebraic approach (an explicit solution that only exists under specific assumptions), these ODEs can also be solved more generally with a numerical scheme (Santos et al., 2018), which we denote as "ODE" in our notations. When NNs are incorporated into this solver, we obtain a neural ODE, denoted "NODE." We have changed this from "NODE" to "UDE" in this revised version for clarify that the NN in the ODE is a state-dependent NN and used to approximate or correct several source terms in the physical ODE system (in addition, the system is solved using an implicit numerical scheme). This is one of the key novelty of our work and has been clarified in the subsections 1.2 and 1.3 of the introduction section. The notation after the dot specifies the type of regionalization mapping used to estimate model parameters, for example, "MLP" in "ODE.MLP." We have explained this point in Sect. 3.1 and added a schematic view of the methods (Fig. 3) in addition to Table 1 for clarification.

RC: Include at least one subsection about neural networks, such as MLPs or CNNs. Most importantly, I still do not know what the input variables are for both NNs. Even though they are not complex neural networks, please write about their basic architecture and hyperparameters. I know some information is already in section 2.3. Please refine it and make it easier to see, such as by adding a small table, rather than hiding it within lines.

AR: Yes, we totally agree with this remark, all details on NN components have been added in Appendix A now.

RC: Although this paper focuses on hybrid models, comparing them to a pure-ML baseline would be beneficial. If it requires too much work, including references to give readers a concept of the accuracy of pure-ML models in simulating river runoffs would also be valuable.

AR: Thank you for this suggestion. We would first like to emphasize that the scope of this paper is the development of a new and more generic hybrid modeling approach (enabling to solve ODE embedding neural networks depending on states), building upon approaches that we have already rigorously tested on large-sample

datasets. Our primary focus here is on resolving mathematical and numerical challenges in hybrid modeling, as well as ensuring physical interpretability, rather than on performance benchmarking. We agree that a more comprehensive evaluation of model performance, including comparisons with pure ML approaches on larger datasets, is important. We greatly appreciate this suggestion and have already planned such a study, which will be carried out shortly.

> Beyond the current evaluation, future work will include explicit benchmarking of the proposed hybrid physics–AI methods against classical conceptual hydrological models as well as pure DL models, such as LSTM-based rainfall–runoff models (e.g., Kratzert et al., 2018). Such comparisons will help quantify the relative benefits of embedding physical constraints versus fully data-driven learning.

Finally, we agree that at least including references to give readers a concept of the accuracy of pure-ML models in simulating river runoffs is valuable.

> These purely data-driven models have been applied successfully to hydrological prediction, achieving state-of-the-art performance in various applications using long short-term memory (LSTM) network (Kratzert et al., 2018; Feng et al., 2020; Cho and Kim, 2022) and their variants like LSTM-based Seq2Seq model (Xiang et al., 2020). For example, Kratzert et al. (2018) reported that across 241 catchments in the U.S., their LSTM model achieved a mean Nash-Sutcliffe efficiency (NSE) of 0.63 in temporal validation, with over 50% of catchments reaching NSE values above 0.65.

RC:   The authors have shown the horizontal resolution is 500m or 1km, and a time step of 1 hour. Could they also state how many grid cells are in the region for calibration? Also how much of the GPU/CPU time are used for GRs and NODEs? Will adding the neural net components significantly add to the computational burden of the host model? In Newton iterations, please show the convergence tolerances and the usual iteration steps.

AR:   Thank you for this comment. We have added all details about the size of the study zone, computational time, convergence criterion, etc. in the revised manuscript (see Sect. 3.1, 2.2 and Appendix B).

RC:   It is better to add multi-basin tests (at least one contrasting basin) to demonstrate the generalization capability and robustness of the NN parameters and the NODE (now denoted UDE) system.

AR:   Thank you for this suggestion. While we agree that testing on multiple contrasting basins can further demonstrate the generalization capability of our methods, we emphasize that the Aude basin test case is already a medium-sized, multi-catchment setup (25 catchments) representing diverse hydrological conditions. We believe that this area provides a sufficiently complex and realistic scenario to demonstrate the proposed algorithms and their methodological innovations, particularly for a software-focused journal like GMD.
As highlighted in previous responses, the primary goal of this paper is the development and rigorous testing of the hybrid modeling framework, including the implicit solver for neural ODE and NN integration, rather than comprehensive performance benchmarking. Extensive generalization tests across multiple basins and comparisons with pure ML and hybrid approaches are planned and will be addressed in a dedicated follow-up study, which will require significant computational effort, especially when incorporating a coupled 1D hydrologic–hydraulic modeling chain.

RC:   Are the physical budgets constrained? For example, water conservation. In the runoff scenario, it would be storage = rainfall – ET – runoff. So, plot the cumulative rainfall – ET – storage – runoff closure and show how the NNs affect them.

AR:   Thank you for this remark. The physical constraints are ensured by the ODE system underlying the hydrological model (with reservoir storage), which could be replaced by other physical laws within the proposed framework. Mass conservation using the same model structure—except for the ODE solver, which is one of the key novelties presented here—has been investigated in detail in Huynh et al. (2025) over a large sample. We have clarified this in Sect. 2.1 and the conclusions of the revised manuscript.

The physical constraints are enforced by the UDE system that underlies the hydrological state-space model and can be flexibly replaced by alternative physical laws within the proposed framework. Note that mass conservation and non-conservative exchange fluxes have been further investigated and analyzed over a large sample using an algebraic resolution of the ODE system in Huynh et al. (2025). The closure relation in Equation 2 follows a simple flux summation under the GR-like hypothesis at each pixel (for a detailed algebraic formulation, see Colleoni et al. (2025) and Huynh et al. (2025)).

(...) future analyses will explore water budget assessments using satellite-derived products to evaluate the realism of the learned flux corrections.

**2.2. Minor comments**

RC: Line 14: Check the font style of "smash." Should it be capitalized or enclosed in quotation marks?

AR: Thank you for pointing this out. We have checked the formatting and now consistently use `smash` throughout the manuscript.

RC: Line 116: What does "neutralized" mean here for precipitation and evaporation?

AR: We refer to net rainfall or evapotranspiration, which is neutralized by the interception reservoir. This terminology is specific to GR models, as referenced in Perrin et al. (2003) and Santos et al. (2018). We have added this clarification to the revised manuscript.

RC: Line 120: "The neural network $\phi$ takes the model states as part of its inputs, thus affecting the model dynamics and state gradient information. It is expected to learn the model behavior by leveraging memory effects through state updates." I do not know how the memory effects is learned by the neural network? Please explain.

AR: Thank you for this comment. The memory effects arise because the neural network $\phi$ takes as input the model states and is embedded in the ODE system to refine internal water fluxes (corrections of source terms in the right-hand side of the ODEs), forming a UDE. These UDEs with state-dependent neural networks implicitly encode the system's memory of past forcings and responses; the dependence becomes more explicit during numerical resolution, where the previous state is used as the initial value for each time step. This have been clarified in the introduction, methodology, and conclusion sections of the revised manuscript.

RC: Figure 2. Please use the specific field names in the caption instead of

AR: Thank you for this comment. We have revised the figure to include notation and descriptor name with their description given in the caption.

RC: Figure 6. Please flip the histogram of precipitation. An inverted view is difficult to interpret. Additionally, it would be better to combine the left three panels into one, increase their heights, and do the same for the right panels. This will make it easier to compare different models.

AR: Thank you for this comment. The figure is corrected as suggested.

RC: Section 4. This section is not appropriate for this paper, which is neither a review nor an opinion piece on physics-guided AI. It should be condensed into a paragraph and added to the conclusion section.

AR: Thank you for this comment. While the paper is not intended as a review, we chose to keep Section 4 to share a broader perspective on pure AI and hydrology. As our work stands at the interface of these two domains, we believe this mixed understanding is valuable and not commonly represented. The section aims to situate our contribution in a wider context and to encourage cross-disciplinary dialogue.

**References**

Cho, K., Kim, Y., 2022. Improving streamflow prediction in the wrf-hydro model with lstm networks. Journal of Hydrology 605, 127297. doi:.

Clark, M.P., Kavetski, D., 2010. Ancient numerical daemons of conceptual hydrological modeling: 1. fidelity and efficiency of time stepping schemes. Water Resources Research 46. doi:.

Colleoni, F., Huynh, N.N.T., Garambois, P.A., Jay-Allemand, M., Organde, D., Renard, B., De Fournas, T., El Baz, A., Demargne, J., Javelle, P., 2025. Smash v1.0: a differentiable and regionalizable high-resolution hydrological modeling and data assimilation framework. Geoscientific Model Development 18, 7003–7034. doi:.

Feng, D., Fang, K., Shen, C., 2020. Enhancing streamflow forecast and extracting insights using long-short term memory networks with data integration at continental scales. Water Resources Research 56, e2019WR026793. doi:.

Huynh, N.N.T., Garambois, P.A., Colleoni, F., Renard, B., Roux, H., Demargne, J., Jay-Allemand, M., Javelle, P., 2024. Learning regionalization using accurate spatial cost gradients within a differentiable high-resolution hydrological model: Application to the french mediterranean region. Water Resources Research 60, e2024WR037544. doi:.

Huynh, N.N.T., Garambois, P.A., Renard, B., Colleoni, F., Monnier, J., Roux, H., 2025. A distributed hybrid physics–ai framework for learning corrections of internal hydrological fluxes and enhancing high-resolution regionalized flood modeling. Hydrology and Earth System Sciences 29, 3589–3613. doi:.

Kratzert, F., Klotz, D., Brenner, C., Schulz, K., Herrnegger, M., 2018. Rainfall–runoff modelling using long short-term memory (lstm) networks. Hydrology and Earth System Sciences 22, 6005–6022. doi:.

Perrin, C., Michel, C., Andrèassian, V., 2003. Improvement of a parsimonious model for streamflow simulation. Journal of hydrology 279, 275–289. doi:.

Santos, L., Thirel, G., Perrin, C., 2018. Continuous state-space representation of a bucket-type rainfall-runoff model: a case study with the gr4 model using state-space gr4 (version 1.0). Geoscientific Model Development 11, 1591–1605. doi:.

Song, Y., Knoben, W.J.M., Clark, M.P., Feng, D., Lawson, K., Sawadekar, K., Shen, C., 2024. When ancient numerical demons meet physics-informed machine learning: adjoint-based gradients for implicit differentiable modeling. Hydrology and Earth System Sciences 28, 3051–3077. doi:.

Xiang, Z., Yan, J., Demir, I., 2020. A rainfall-runoff model with lstm-based sequence-to-sequence learning. Water Resources Research 56, e2019WR025326. doi:.